# 👠 SyntheRela: A Benchmark For Synthetic Relational Database Generation

**Valter Hudovernik**$^*$**, Martin Jurkovič**$^*$**, Erik Štrumbelj**$^\dagger$
*Faculty of Computer and Information Science, University of Ljubljana*
$^\dagger$*Corresponding author: erik.strumbelj@fri.uni-lj.si*

Reviewed on OpenReview: *https://openreview.net/forum?id=Mi8XioazWy*

## Abstract

Synthesizing relational databases has started to receive more attention from researchers, practitioners, and industry. The task is more difficult than synthesizing a single table due to the added complexity of relationships between tables. For the same reasons, benchmarking methods for synthesizing relational databases introduces new challenges. Our work is motivated by a lack of an empirical evaluation of state-of-the-art methods and by gaps in the understanding of how such an evaluation should be done. We review related work on relational database synthesis, common benchmarking datasets, and approaches to measuring the fidelity and utility of synthetic data. We combine best practices, a novel robust detection metric, and a novel approach to evaluating utility with graph neural networks into a benchmarking tool. We use this benchmark to compare 6 open-source methods over 8 real-world databases, with a total of 39 tables. The open-source SYNTHERELA benchmark is available on GitHub with a public leaderboard.

🐙 **Data & Code:** `github.com/martinjurkovic/syntherela`
🤗 **Leaderboard:** `huggingface.co/spaces/SyntheRela/leaderboard`

## 1 Introduction

Synthesizing relational databases - generating relational databases that preserve the characteristics of the original databases - is an emerging field. It promises several benefits, from protecting privacy to addressing data scarcity, while preserving the complexity and dependencies present in the original databases. This makes it attractive for healthcare (Appenzeller et al., 2022), finance (Assefa et al., 2020), and education (Bonnéry et al., 2019), where accessing and utilizing data can be challenging due to privacy concerns, data scarcity, or biases (Ntoutsi et al., 2020; Rajpurkar et al., 2022).

The foundations of synthesizing relational databases were laid by the Synthetic Data Vault (Patki et al., 2016). Recently, several deep learning methods have been proposed (Gueye et al., 2023; Li et al., 2024; Mami et al., 2022; Xu et al., 2023; Canale et al., 2022; Solatorio & Dupriez, 2023; Pang et al., 2024; Hudovernik, 2024). The field has also received attention from the industry, with several commercial tools now available and with Google, Amazon, and Microsoft integrating them into their cloud services (Gretel.ai, 2024).

Although there are several packages for evaluating the quality of synthetic tabular data, only the SDMetrics package (Patki et al., 2016) provides some support for the evaluation of synthetic relational databases. As such, the field lacks not only an empirical comparison of available methods but also an understanding of how such an evaluation should be done. We address this gap with an evaluation methodology that combines established evaluation metrics (Section 2.2), best practices, sampling procedures, and real-world relational databases (Table 1). We also propose a **detection-based metric C2ST-Agg** (Section 3.1) specialized for

---

$^*$Equal contribution.

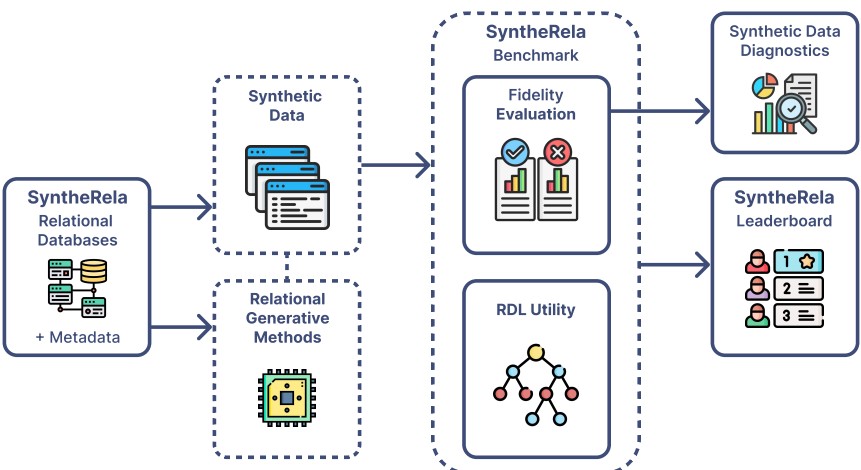

Figure 1: **SYNTHERELA** enables the evaluation of synthetic relational database generation methods using real-world relational databases. SYNTHERELA supports the evaluation of fidelity using established metrics and a relational detection-based metric, C2ST-Agg. It also assesses utility via relational deep learning (RDL) utility. The benchmark provides synthetic data diagnostics using ML explainability and a leaderboard for comparing method performance.

relational databases and use ML explainability to diagnose issues with synthetic data generation methods, and a novel approach to evaluate the utility of synthetic relational databases with **relational deep learning utility** (Section 3.2).

We implement the methodology in **SYNTHERELA**, a benchmark and evaluation tool that is available as an open source package and can be easily extended with new metrics and datasets (see Appendix B). Finally, we use the benchmark to evaluate current state-of-the-art methods (Section 2.1) over several relational databases. This is the first comprehensive evaluation and comparison of methods for the synthesis of relational databases and provides valuable insights into their ability to synthesize relational aspects of the data (Section 4). We release the code of our benchmark under an open source license alongside a public leaderboard.

## 2 Related Work

The quality of synthetic data consists of three core aspects: *fidelity*, *utility* and *privacy* (Snoke et al., 2018; Beaulieu-Jones et al., 2019; Qian et al., 2023a). **Fidelity** captures how much the synthetic data resembles the statistical properties of the real data. **Utility** reflects how well the synthetic data can replace the original in downstream tasks. **Privacy** measures the degree of protection of the identities of the original subjects. It is typically defined through the theoretical framework of differential privacy (DP) (Dwork, 2006) or the notion of membership disclosure (El Emam et al., 2022a). Our evaluation is focused on the quality of synthetic data (fidelity and utility) while ensuring it does not compromise the privacy of the original subjects.

### 2.1 Methods for Synthesizing Relational Databases

In this work, we focus on relational databases — a collection of tables linked by primary and foreign keys. We distinguish this from synthesizing tabular data (a single-table), which is a special case and an even more active field (Borisov et al., 2022; Qian et al., 2023b). Existing methodologies for generating synthetic relational data can be broadly categorized into two main approaches: neural network–based methods and marginal-based methods. We provide a concise summary of each below.

Patki et al. (2016) are the first to propose a method for relational database synthesis - The Synthetic Data Vault (SDV), which employs Gaussian copulas with a hierarchical modeling algorithm and does not fit into either of these two main categories.

Neural network–based methods aim to preserve data fidelity and utility across arbitrary schemas. Row Conditional-TGAN (RCTGAN) (Gueye et al., 2023) and Incremental Relational Generator (IRG) (Li et al., 2024) are based on GANs. The Realistic Relational and Tabular Transformer (REaLTabFormer) (Solatorio & Dupriez, 2023) and Composite Generative Models (Canale et al., 2022) are based on transformers. The work of Mami et al. (2022) is based on graph variational autoencoders, while Xu et al. (2023) propose a framework for synthesizing many-to-many datasets using random graphs. Pang et al. (2024) propose ClavaDDPM, a method based on classifier-guided diffusion models. Hudovernik (2024) proposes RGCLD, a method based on graph neural networks (GNNs) and conditional diffusion models. Recently, Tiwald et al. (2025) propose TabularARGN - a framework for tabular and relational data synthesis based on autoregressive tabular models.

Marginal-based methods, by contrast, involve taking low-dimensional measurements of the dataset, such as marginal queries, and adding calibrated statistical noise to ensure DP. While well established in the single-table setting (Zhang et al., 2017; McKenna et al., 2022; Cai et al., 2021), extensions to relational data are more recent. PrivLava (Cai et al., 2023) models inter-table dependencies using graphical models with latent variables. MARE (Kapenekakis et al., 2024) targets medical relational datasets, such as electronic health records, with temporal and sequential characteristics. More recently, Alimohammadi et al. (2025) proposed a method for adapting single-table DP generators to relational data which models foreign keys by learning a bi-adjacency matrix. Finally, PrivPetal(Cai et al., 2025) synthesizes relational data by modeling flattening relational databases in an efficient manner. These approaches generally prioritize privacy and query accuracy over flexibility and general fidelity.

Our evaluation in this work will focus on neural network-based methods for the generation of synthetic relational data. This focus is driven by the fact that their implementations are generally applicable to generating synthetic data for arbitrary relational schemas. In contrast, marginal-based methods often include implementations tailored towards specific datasets with predefined query workloads. We provide a more detailed overview of related work in Appendix A.

## 2.2 Metrics for Evaluating Synthetic Data

The quality of synthetic tabular data and relational databases is evaluated primarily based on two criteria, *fidelity* and *utility*. Fidelity measures the degree of similarity between synthetic and real data in terms of its properties, while utility measures how well synthetic data can be used in place of real data when the data are part of a downstream task, for example, predictive modeling (Hansen et al., 2023). The utility of synthetic data is typically assessed with the train-on-synthetic evaluate-on-real methods (Beaulieu-Jones et al., 2019). We further divide fidelity metrics into *statistical*, *distance-based*, and *detection-based* metrics.

**Statistical fidelity** methods are typically used to assess marginal distributions, sometimes bivariate distributions. The most commonly used methods are the Kolmogorov-Smirnov test and the $\mathcal{X}^2$ test for numerical and categorical variables, respectively. For relational data, cardinality shape similarity is used, where for each parent row the number of child rows is calculated. This yields a numerical distribution for both real and synthetic data, on which a Kolmogorov-Smirnov test is performed.

Similar to statistical fidelity, **distance-based fidelity** is typically used to assess the quality of marginal distributions. However, some distance metrics also assess entire tables. Commonly used distance-based methods are total variation distance, Kullback-Leibler divergence, Jensen-Shannon distance, Wasserstein distance, maximum mean discrepancy, and pairwise correlation difference. To evaluate inter-table relationships Pang et al. (2024) use $k$-hop similarity, where they compute correlations between tables at distance $k$ (0-hop refers to columns within the same table, 1-hop refers to columns in tables directly connected via a foreign key, etc.). Unlike statistical methods, reports of distance-based fidelity do not include hypothesis testing or any other quantification of uncertainty. This is an issue both when evaluating a method and when comparing two methods. In the former, a method can achieve a seemingly high distance that is in a high probability region when taking into account the sampling distribution. In the latter, a seemingly large difference between the two methods can be explained away by the variance of the sampling distribution.

The basic idea of **detection-based fidelity** is to learn a model that can discriminate between real and synthetic data. The detection-based metric can be interpreted as a null-hypothesis test for comparing two distributions (two sample testing) with classification accuracy as a proxy Kim et al. (2021). The classifier

serves as a map from high-dimensional data to a one-dimensional test statistic. In machine learning literature, this is referred to as a classifier two sample test (C2ST) (Lopez-Paz & Oquab, 2017). If the model can achieve better-than-random predictive performance, this indicates that there are some patterns that identify synthetic data. Zein & Urvoy (2022) show that using appropriate discriminative models can highlight the differences between real and synthetic tabular data.

The most common detection-based metric is logistic detection (LD) (Gueye et al., 2023; Solatorio & Dupriez, 2023; Li et al., 2024; Pang et al., 2024), where a logistic regression model is used for discrimination. An extended version of LD known as parent-child logistic detection (P-C LD) is used to evaluate relational databases. P-C LD applies LD to denormalized pairs of synthetic parent and child tables, assessing the preservation of parent-child relationships. A serious issue with denormalization is that it may introduce correlation between rows, breaking the *i.i.d.* assumption. This results in an over-performance of the discriminative model and in underestimating the quality of the method for synthesizing relational data. It also makes it impossible to set a detection threshold for testing fidelity (for example, accuracy would be greater than 50% even if both datasets were from the same data generating process). For these reasons, we do not consider P-C detection.

Note that logistic regression is unable to capture interactions between columns unless these interactions are explicitly included as features. This implies a lenient evaluation of the state-of-the-art methods (we demonstrate this empirically in Appendix C.3). Tree-based ensemble models are a better alternative, which is also suggested by the findings of Zein & Urvoy (2022) for tabular data.

The **utility** of synthetic data is most commonly measured with machine learning efficacy - comparing the hold-out performance of a predictive model trained on the original data with a predictive model trained on synthetic data (Canale et al., 2022; Li et al., 2024; Mami et al., 2022; Solatorio & Dupriez, 2023; Pang et al., 2024). Patki et al. (2016) measured utility with a user study and Hansen et al. (2023) with the ability to retain model or feature importance ranking (measured with rank correlation) in the train-on-synthetic evaluate-on-real paradigm. Note that all of these studies evaluated utility on a single-table, even those that investigated synthetic relational databases.

## 3 Evaluating Synthetic Relational Databases

### 3.1 Multi-Table Fidelity Using Aggregation

We build our approach on existing detection-based approaches. First, we address the lenient evaluation by replacing logistic regression commonly used in related work with tree-based ensemble methods (Borisov et al., 2023; Zein & Urvoy, 2022).

Current fidelity metrics fail to thoroughly evaluate interactions induced by relationships between tables: (i) cardinality similarity [1] assesses only the cardinality of foreign key relationships, (ii) $k$-hop similarity (Pang et al., 2024) evaluates only linear relationships between column pairs in related tables, and (iii) existing implementations of PC-LD [2] do not properly account for denormalization breaking the *i.i.d.* assumption required for C2ST, leading to unprincipled inference.

To address these limitations, we introduce a classifier two-sample test with aggregation (C2ST-Agg). Instead of denormalizing tables, we incorporate the multi-table structure by augmenting both real and synthetic parent tables with aggregated features derived from their child tables.

Aggregation is an established technique in the field of relational reasoning (Getoor et al., 2007; Džeroski, 2010) and C2ST-Agg can be thought of as a propositionalization (Kramer et al., 2001) approach to the C2ST on relational databases. By summarizing child-table columns and relationship cardinality through aggregation functions (e.g., mean, count, max), C2ST-Agg maintains the *i.i.d.* assumption while enhancing classifier-based fidelity assessment.

---

[1] https://docs.sdv.dev/sdmetrics/data-metrics/quality/cardinalityshapesimilarity
[2] https://github.com/sdv-dev/SDMetrics/blob/00f63b65bca4d57eee08018e499f3a634399ff65/sdmetrics/multi_table/detection/parent_child.py

Our approach addresses the issues of current fidelity metrics: it accounts both for relationship cardinality (i) and high-level interactions across all columns in related tables (ii), while maintaining the *i.i.d.* assumption for each table (iii).

In practice, users can select aggregation functions based on the specific aspects of relational data they wish to evaluate. We provide general guidelines for choosing these aggregations, along with a detailed explanation of the C2ST-Agg metric, in Appendix B.2.1

A benefit of using predictive models to evaluate fidelity is that we can use ML explainability methods to diagnose issues with our data generation. By examining the features that the discriminative model uses to distinguish between synthetic and real data, we can identify which aspects of the data the generative method fails to model well. We can employ standard ML interpretability methods such as Shapley values (Strumbelj & Kononenko, 2010; Lundberg & Lee, 2017), partial dependence plots (Friedman, 2001), accumulated local effects (Apley & Zhu, 2020), or model-specific methods such as relative variable importance in boosted trees or coefficients of linear regression models. See Section 4.2 for an example.

### 3.2 Relational Deep Learning Utility

Relational deep learning (Fey et al., 2024) has emerged as an alternative to traditional ML methods by transforming relational databases into graphs (Robinson et al., 2024; Papamarkou et al., 2024) and utilizing graph neural networks (GNNs). Subsequently, Robinson et al. (2024) proposed RelBench, a benchmark for relational deep learning. The authors of RelBench show that the performance of a GNN pipeline is comparable to a traditional ML pipeline approach done by a data scientist.

Up until now, ML utility has been evaluated only on single-tables (see Section 2.2), which means inter-table relationships have not been taken into account even when evaluating multi-table synthetic data. We could create a feature engineering pipeline for each database, but that would require domain knowledge, is labor-intensive, and is not easily extensible. Instead, we incorporate the graph-based approach of RelBench into SYNTHERELA. We adopt the RelBench pipeline to fit GNN models on real and synthetic data respectively, and then evaluate them on the test set consisting entirely of real data (see Appendix D for more details).

Splitting the data into train and test sets is challenging for relational databases with complex inter-table dependencies. To address this, we adopt a time-based splitting strategy, consistent with the methodology proposed by RelBench. This allows us to evaluate data generation performance for any relational database with a time component, without the need for manual data processing. It can also be easily extended to new databases, provided they contain a datetime column allowing a train-test split. Notably, this is the only approach to relational utility that includes the entire relational database, ensuring a more comprehensive evaluation. We call this approach **relational deep learning utility (RDL-utility)**.

## 4 Benchmarking and Results

We combine our findings into a synthetic relational database benchmark. We report existing metrics for evaluating single column, single-table, and multi-table fidelity to which we add a new metric for measuring multi-table fidelity C2ST-Agg and a novel approach to evaluating the utility of relational synthetic data, RDL-utility.

We compare the following methods for synthesizing relational data: **SDV**, **RCTGAN**, **REaLTabFormer**, **ClavaDDPM**, **RGCLD**, and **TabularARGN**. Other related work does not have available source code, does not have an API or we were not able to run the source code on the selected databases.

We include 6 datasets featured in related work (**Airbnb**, **Rossmann**, **Walmart**, **Biodegradability**, **MovieLens**). In addition we include **Cora** dataset by McCallum et al. (2000), a popular dataset in graph representation learning, and the **F1** dataset from the relational deep learning benchmark Robinson et al. (2024), for a total of **8 benchmark datasets**. The datasets vary in types of relationships and number of tables and columns, which are summarized in Table 1 (see Appendix B.3 for details and Appendix G for database schemas).

Table 1: **Summary of the 8 benchmark datasets.** The number of columns represents the number of non-id columns. The collection is diverse and covers all types of relational structures.

| Dataset Name | # Tables | # Rows | # Columns | # Relationships | Hierarchy Type |
|---|---|---|---|---|---|
| Rossmann | 2 | 59,085 | 16 | 1 | Linear |
| Airbnb | 2 | 57,217 | 20 | 1 | Linear |
| Walmart | 3 | 15,317 | 17 | 2 | Multi Child |
| Cora | 3 | 57,353 | 2 | 3 | Multi Child |
| Biodegradability | 5 | 21,895 | 6 | 5 | Multi Child & Parent |
| IMDB MovieLens | 7 | 1,249,411 | 14 | 6 | Multi Child & Parent |
| Berka | 8 | 757,722 | 37 | 8 | Multi Child & Parent |
| F1 | 9 | 74,063 | 33 | 13 | Multi Child & Parent |

We evaluate all three levels of synthetic relational databases, with a focus on multi-table evaluation (see Appendix B.2 for all benchmark metrics). Most methods are non-deterministic, so we report results for three different replications. However, all results are stable across replications. Four of the methods are capable of synthesizing all of the datasets, irrespective of their relational structure. REALTABFORMER is only capable of generating databases with linear structure and ClavaDDPM is unable to model datasets with two or more foreign keys between a pair of tables (*Biodegradability* and *CORA*).

For single-column metrics (see Table 12 in the Appendix for results), we report the complement of the Kolmogorov-Smirnov statistic and the Total Variation Distance (the complement to KS/TV distance between two distributions P and Q is $1 - D_{\text{KS/TV}}(P||Q)$). For single-table metrics, we report the complements of the KS and TV distances between column pair correlations. For multi-table metrics, we report the average cardinality shape similarity and the $k$-hop ($k > 1$) correlations between tables. At all levels, we report the C2ST using XGBoost (Chen & Guestrin, 2016) as the discriminative model. We use 5-fold stratified cross-validation to estimate detection accuracy. For C2ST-Agg, we augment the rows with (a) counts of child rows for each row in each parent table, (b) the mean values of the numeric columns in the child table corresponding to the parent row; and (c) the number of unique categories in related rows.

Neural-based single-table generators evaluate privacy with metrics that use a held-out test set, as opposed to marginal-based generators, that provide provable privacy guarantees (DP). As representative sampling for relational databases is non-trivial, evaluating privacy in multi-table data remains an open problem. Recently, the MIDST Challenge (Vector Institute, 2025; Wu et al., 2025) provided the first analysis of the privacy of neural-based relational data generators, focusing on membership inference attacks (MIA) (Shokri et al., 2017). These attacks are tailored to individual generative models and are therefore not generally applicable. Instead, we performed a privacy sanity check against the SMOTE baseline, following related work (Pang et al., 2024; Tiwald et al., 2025). We use the *distance to closest record* metric and perform our analysis on the *Airbnb* dataset as we can re-sample a hold-out set of equal size without a temporal shift. The results of the privacy check show that all the methods outperform SMOTE, providing no evidence of systematic data copying (see Appendix C.1). We also explore how C2ST can be used as a data copying diagnostic in Appendix C.4.

## 4.1 Single-Table Performance

We first evaluate the fidelity of individual tables. We focus on the detection metric and how well column pairs (bivariate distributions) are modeled. Table 2 summarizes the results. On the databases that it is able to generate, the diffusion-based ClavaDDPM performs best, followed by the autoregressive TabularARGN specializing in sequential (linear structured) databases.

The rankings of methods for single column metrics are similar to those of single-tables. As expected, the methods model individual columns better. See Appendix C.2 for details. Interestingly, TabularARGN performs best on modeling marginal distributions, indicating their discretization approach is a robust preprocessing step.

Table 2: **Single-table results**. For each dataset and metric we report the average and standard error of C2ST accuracy (lower is better) and column pair trends (Pairs - higher is better) across all tables for three independent samples. The best result for each dataset is **in bold**.

| Dataset | Metric | TARGN | RGCLD | ClavaDDPM | RCTGAN | REALTABF. | SDV |
|---|---|---|---|---|---|---|---|
| Airbnb | C2ST ($\downarrow$) | **64.23**$_{\pm0.20}$ | 70.25$_{\pm3.31}$ | 78.10$_{\pm0.03}$ | 88.37$_{\pm0.14}$ | 83.97$_{\pm4.36}$ | 99.75$_{\pm5e\text{-}3}$ |
| | Pairs ($\uparrow$) | **93.48**$_{\pm0.33}$ | 88.51$_{\pm0.60}$ | 87.78$_{\pm0.12}$ | 79.37$_{\pm0.29}$ | 53.90$_{\pm1.26}$ | 49.03$_{\pm0.08}$ |
| Rossmann | C2ST ($\downarrow$) | **56.07**$_{\pm0.58}$ | 68.97$_{\pm3.85}$ | 66.77$_{\pm0.14}$ | 88.02$_{\pm0.50}$ | 74.70$_{\pm0.55}$ | 96.90$_{\pm0.21}$ |
| | Pairs ($\uparrow$) | **91.34**$_{\pm0.08}$ | 90.08$_{\pm0.58}$ | 84.78$_{\pm0.80}$ | 84.38$_{\pm0.40}$ | 84.58$_{\pm0.88}$ | 67.77$_{\pm0.25}$ |
| Walmart | C2ST ($\downarrow$) | 83.54$_{\pm0.84}$ | 66.76$_{\pm1.82}$ | **53.50**$_{\pm1.95}$ | 76.40$_{\pm0.55}$ | 70.87$_{\pm1.07}$ | 87.02$_{\pm0.81}$ |
| | Pairs ($\uparrow$) | 83.89$_{\pm0.21}$ | 91.74$_{\pm0.99}$ | **94.02**$_{\pm0.14}$ | 86.60$_{\pm0.25}$ | 83.10$_{\pm0.46}$ | 87.61$_{\pm0.23}$ |
| Berka | C2ST ($\downarrow$) | 72.31$_{\pm0.17}$ | 64.17$_{\pm2.04}$ | **54.48**$_{\pm0.11}$ | 68.12$_{\pm0.44}$ | - | 82.40$_{\pm0.33}$ |
| | Pairs ($\uparrow$) | 70.43$_{\pm0.25}$ | 73.77$_{\pm1.75}$ | **88.54**$_{\pm1.19}$ | 74.22$_{\pm0.28}$ | | 64.01$_{\pm0.11}$ |
| F1 | C2ST ($\downarrow$) | 81.93$_{\pm0.49}$ | **69.63**$_{\pm1.23}$ | 71.42$_{\pm0.46}$ | 80.67$_{\pm0.31}$ | - | 89.84$_{\pm0.22}$ |
| | Pairs ($\uparrow$) | 81.31$_{\pm0.45}$ | **92.39**$_{\pm0.77}$ | 84.65$_{\pm0.05}$ | 90.17$_{\pm0.03}$ | | 73.05$_{\pm0.19}$ |
| IMDB | C2ST ($\downarrow$) | 50.92$_{\pm0.22}$ | 55.01$_{\pm2.36}$ | **49.83**$_{\pm0.07}$ | 55.38$_{\pm0.11}$ | - | TLE |
| | Pairs ($\uparrow$) | 97.80$_{\pm0.13}$ | 93.89$_{\pm2.42}$ | **98.66**$_{\pm0.10}$ | 81.65$_{\pm0.03}$ | | |
| Biodeg. | C2ST ($\downarrow$) | 58.79$_{\pm0.19}$ | 63.45$_{\pm2.26}$ | - | **58.26**$_{\pm0.15}$ | - | 68.59$_{\pm0.14}$ |
| | Pairs ($\uparrow$) | 74.82$_{\pm0.35}$ | 90.89$_{\pm5.05}$ | | 85.44$_{\pm2.19}$ | | **97.58**$_{\pm0.50}$ |
| Cora | C2ST ($\downarrow$) | 50.94$_{\pm0.37}$ | 52.98$_{\pm1.03}$ | - | **48.97**$_{\pm0.14}$ | - | 75.45$_{\pm0.16}$ |

## 4.2 Multi-Table Performance

Multi-table metrics examine how well the relationship cardinality and the relationships between different tables are preserved. Cardinality shape similarity examines only the former, while $k$-HOP similarity evaluates the latter; C2ST-Agg examines both. We report the results in Table 3. Similar to single-table fidelity, the diffusion-based methods perform best.

C2ST with aggregation uses the same features for single-table and multi-table evaluation (with the exception of the aggregation attributes). This allows us to directly compare single and multi-table performance. For most methods, we observe a significant drop in fidelity when adding aggregations. Figure 2 shows how C2ST detection accuracy increases when we add information about relationships between tables. We investigate this further using explainability methods and show how these can be used to debug generative methods.

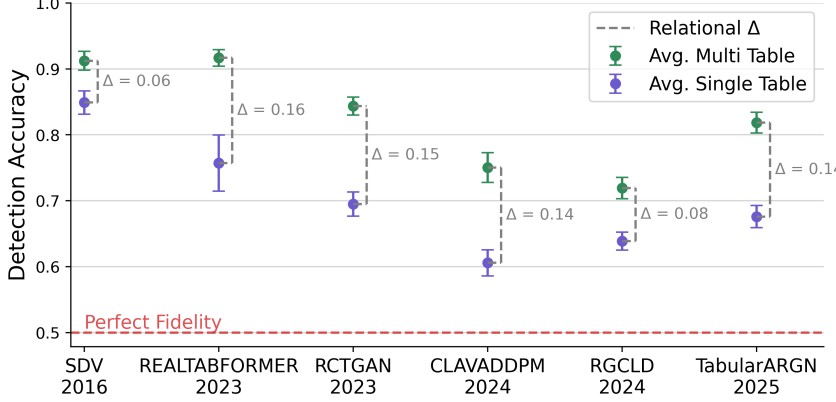

Figure 2: **Comparing single and multi-table performance**. While there is an overall trend of improvement over time of both single-table and multi-table fidelity, most methods still exhibit a significant gap between single-table and multi-table fidelity (relational $\Delta$).

Table 3: **Multi-table results.** For each dataset we report the average and standard error of C2ST-Agg accuracy (lower is better), cardinality similarity (higher is better) and k-hop correlation similarity (higher is better) across all tables for three independent samples. "-" denotes a method is unable to generate the dataset, and TLE timeout. The best result is in **bold** and results within one standard deviation are underlined.

| Dataset | Metric | TARGN | RGCLD | ClavaDDPM | RCTGAN | REALTABF. | SDV |
|---|---|---|---|---|---|---|---|
| Airbnb | C2ST-Agg (↓) | **$63.47_{\pm0.88}$** | $79.53_{\pm5.39}$ | $\approx 100.0$ | $98.22_{\pm0.08}$ | $99.13_{\pm0.02}$ | $99.94_{\pm0.01}$ |
| | Cardinality (↑) | $98.59_{\pm0.32}$ | $99.36_{\pm0.10}$ | **$99.65_{\pm0.06}$** | $95.45_{\pm0.62}$ | $76.38_{\pm0.47}$ | $26.36_{\pm0.03}$ |
| | 1-HOP (↑) | $79.66_{\pm0.36}$ | $84.50_{\pm2.74}$ | **$86.69_{\pm0.14}$** | $68.78_{\pm0.54}$ | $33.99_{\pm5.76}$ | $24.58_{\pm0.03}$ |
| Rossmann | C2ST-Agg (↓) | **$60.43_{\pm0.63}$** | $75.56_{\pm0.74}$ | $85.77_{\pm0.07}$ | $86.11_{\pm1.01}$ | $85.90_{\pm1.33}$ | $98.37_{\pm0.23}$ |
| | Cardinality (↑) | $94.17_{\pm1.84}$ | $98.95_{\pm0.50}$ | **$99.19_{\pm0.29}$** | $82.69_{\pm1.95}$ | $41.82_{\pm10.29}$ | $99.16_{\pm0.15}$ |
| | 1-HOP (↑) | **$92.95_{\pm0.78}$** | $88.30_{\pm0.38}$ | $82.81_{\pm0.47}$ | $87.02_{\pm0.17}$ | $80.25_{\pm0.84}$ | $73.84_{\pm0.34}$ |
| Walmart | C2ST-Agg (↓) | $94.81_{\pm1.68}$ | $88.89_{\pm1.05}$ | **$73.33_{\pm2.92}$** | $94.81_{\pm1.68}$ | $90.00_{\pm0.91}$ | $88.52_{\pm1.60}$ |
| | Cardinality (↑) | $65.93_{\pm1.98}$ | **$94.81_{\pm0.30}$** | $93.33_{\pm2.28}$ | $88.15_{\pm1.51}$ | $85.56_{\pm4.57}$ | $86.30_{\pm1.09}$ |
| | 1-HOP (↑) | $75.40_{\pm1.49}$ | $82.05_{\pm2.77}$ | **$86.40_{\pm1.73}$** | $79.02_{\pm0.15}$ | $74.99_{\pm0.20}$ | $76.64_{\pm1.07}$ |
| Berka | C2ST-Agg (↓) | $80.56_{\pm1.86}$ | $72.71_{\pm2.33}$ | **$69.12_{\pm0.63}$** | $76.86_{\pm2.22}$ | | $77.43_{\pm0.14}$ |
| | Cardinality (↑) | $85.17_{\pm0.84}$ | $\approx$ **$100.0$** | $96.43_{\pm0.36}$ | $81.28_{\pm1.07}$ | | $80.53_{\pm0.72}$ |
| | 1-HOP (↑) | $72.82_{\pm0.38}$ | $80.74_{\pm1.63}$ | **$87.92_{\pm1.66}$** | $78.87_{\pm0.91}$ | - | $59.09_{\pm0.49}$ |
| | 2-HOP (↑) | $65.51_{\pm0.31}$ | $73.77_{\pm1.78}$ | **$84.41_{\pm2.46}$** | $77.98_{\pm0.95}$ | | $23.09_{\pm0.21}$ |
| | 3-HOP (↑) | $59.34_{\pm0.62}$ | $64.81_{\pm4.96}$ | **$80.67_{\pm2.18}$** | $78.65_{\pm0.69}$ | | $58.23_{\pm0.58}$ |
| F1 | C2ST-Agg (↓) | $95.90_{\pm0.94}$ | **$74.18_{\pm2.68}$** | $82.52_{\pm0.25}$ | $91.23_{\pm0.39}$ | | $94.55_{\pm0.24}$ |
| | Cardinality (↑) | $58.17_{\pm3.71}$ | $\approx$ **$100.0$** | $88.45_{\pm3.05}$ | $56.82_{\pm1.55}$ | | $71.88_{\pm0.12}$ |
| | 1-HOP (↑) | $77.37_{\pm0.26}$ | **$88.35_{\pm2.01}$** | $79.35_{\pm0.03}$ | $79.14_{\pm0.72}$ | - | $68.45_{\pm0.20}$ |
| | 2-HOP (↑) | $76.25_{\pm0.32}$ | **$87.53_{\pm1.56}$** | $84.18_{\pm0.12}$ | $83.50_{\pm0.82}$ | | $76.93_{\pm0.24}$ |
| IMDB | C2ST-Agg (↓) | $73.76_{\pm1.78}$ | $65.37_{\pm5.54}$ | **$65.00_{\pm0.34}$** | $81.56_{\pm2.00}$ | | |
| | Cardinality (↑) | $81.19_{\pm0.80}$ | $\approx$ **$100.0$** | $98.95_{\pm0.03}$ | $79.53_{\pm1.27}$ | - | TLE |
| | 1-HOP (↑) | $88.64_{\pm0.70}$ | $85.66_{\pm5.25}$ | **$91.57_{\pm1.25}$** | $81.76_{\pm0.20}$ | | |
| Biodeg. | C2ST-Agg (↓) | $88.86_{\pm0.26}$ | **$70.51_{\pm4.08}$** | | $83.82_{\pm3.35}$ | | $98.02_{\pm0.06}$ |
| | Cardinality (↑) | $79.53_{\pm0.24}$ | **$97.95_{\pm0.01}$** | | $85.22_{\pm0.50}$ | | $61.17_{\pm0.36}$ |
| | 1-HOP (↑) | $61.36_{\pm0.47}$ | **$78.46_{\pm4.65}$** | - | $75.80_{\pm1.46}$ | - | $49.09_{\pm0.59}$ |
| | 2-HOP (↑) | $60.54_{\pm0.44}$ | $75.12_{\pm4.02}$ | | **$77.04_{\pm1.96}$** | | $47.80_{\pm2.16}$ |
| Cora | C2ST-Agg (↓) | $68.80_{\pm0.67}$ | **$62.33_{\pm1.60}$** | | $73.74_{\pm0.47}$ | | $99.59_{\pm0.03}$ |
| | Cardinality (↑) | $96.27_{\pm0.13}$ | $\approx$ **$100.0$** | - | $90.48_{\pm2.16}$ | - | $68.82_{\pm0.29}$ |
| | 1-HOP (↑) | **$80.42_{\pm0.34}$** | $63.81_{\pm3.06}$ | | $68.39_{\pm0.08}$ | | $4.95_{\pm0.12}$ |

ML explanation with feature importance reveals that methods struggle with preserving the relationships between columns across tables. Figure 3a shows an example of how aggregation attributes summarizing information about child table rows are the most discriminative features. We further examine two such attributes in Figure 4. The partial dependence plots of the first and fourth most important features from Figure 3 show how subsets of both categorical (Fig. 4a) and numerical (Fig. 4b) features' conditional distributions are informative to the discriminative model. In Figure 3b, we illustrate how a simple interaction between two aggregation attributes—specifically, relationship cardinality and the number of unique categories in a child table—renders the synthetic data nearly separable from the original. This highlights how C2ST-Agg captures both structural and attribute-based discrepancies, including their interactions, offering a more comprehensive evaluation than simpler metrics like k-hop similarity. We include the interpretability methods directly into our implementation of the detection metric, allowing users to immediately investigate where their method is underperforming after evaluating the synthetic data.

## 4.3 Relational Deep Learning Utility Performance

We run extensive experiments for RDL-utility across 5 databases that contain a temporal feature, using 6 GNN architectures with hyperparameter tuning for each architecture and database pair on original data (see Appendix D for more details). Table 4 summarizes the RDL-utility results (Section 3.2). Details of the tasks are provided in the Appendix D. The utility scores follow the same trend as the multi-table results, with models achieving high fidelity also performing best in utility tasks. The best performing models are

the diffusion based approaches ClavaDDPM and RGCLD, with the autoregressive approach TabularARGN performing best on one simpler two-table database. The drop in utility performance for databases with more complex relational structures is also consistent with fidelity results and shows that utility of relational databases should be evaluated using multi-table approaches.

To be able to estimate the value and validity of RDL-utility in measuring the utility performance of the generated synthetic databases, we implement two baseline metrics for measuring utility (see Appendix D.3 for more details). The first one is a simple baseline, where we train a LightGBM (Ke et al., 2017) model directly on the entity table with no feature engineering. This corresponds to the typical machine learning utility evaluation used in related work (Solatorio & Dupriez, 2023; Pang et al., 2024). The second baseline is an automated feature engineering baseline using Deep Feature Synthesis (DFS) (Kanter & Veeramachaneni, 2015), based on calculating aggregated features of the child tables and appending them to the entity table, also trained with a LightGBM model.

Entity table LightGBM results (Table 5) again show that diffusion based models are able to best generate a single table overall. The DFS feature engineering baseline scores (Table 6) show a similar story. The drop in performance in the RDL-utility compared to the entity-table baseline stems from the noise introduced by poorly generating the relational structure and features of child tables.

RDL-utility, alongside naive baselines, demonstrates that the models can learn patterns from synthetic databases that generalize to held-out test sets even in cases of non-perfect fidelity.

### 4.4 Privacy Evaluation

Our primary focus is on evaluating the **fidelity** and **utility** of synthetic relational data. However, high data quality should not come at the expense of **privacy**. Following prior work (Kotelnikov et al., 2023; Pang et al., 2024; Zhang et al., 2024), we conduct a privacy comparison against SMOTE (Chawla et al., 2002), an interpolation-based method that synthesizes new data points via convex combinations of real samples and their nearest neighbors.

To assess privacy risks, we compute the distance to closest record (DCR)(Zhao et al., 2021), a common metric that quantifies how close synthetic records are to real data. Specifically, we report the DCR score—the probability that a synthetic sample is closer to a real training point than to any sample from a held-out test set. We conduct our privacy check on the *Airbnb* dataset containing anonymised information of real users.

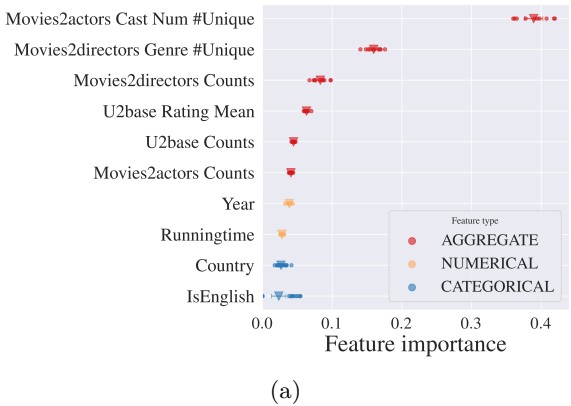

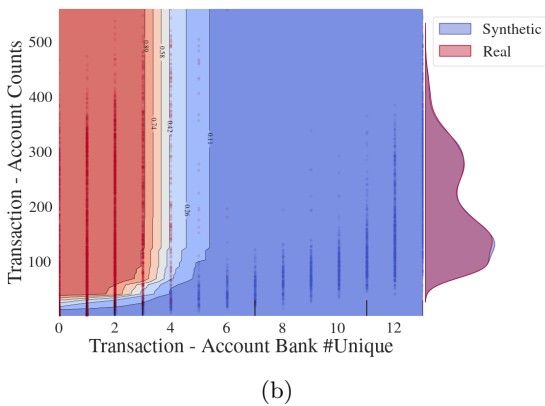

(a)  (b)

Figure 3: **ML Interpretability Diagnostics of C2ST.** (a) Feature importance for C2ST-Agg using XGBoost. Results are for the best-performing method. The added features that incorporate relational information (red) are the most important for discriminating between real and synthetic data. (b) Two-way PDP showing the interaction between the relationship cardinality of the Account–Transaction key (with its marginal distribution shown on the right) and the number of unique banks associated with an account's transactions. The plot illustrates how a combination of structural and informational aggregation attributes can render the synthetic data nearly separable from the original.

Table 4: **RDL-utility results.** We report mean ± SE of ROC-AUC (classification) and MAE (regression). Naive baseline (mean/majority) is in parentheses. "-" indicates invalid time columns. Best is **bold**; within one SE is underlined.

| Dataset | Model | | ORIG. | RGCLD | TARGN | CLAVA | RCTGAN | REALTABF. | SDV |
|---|---|---|---|---|---|---|---|---|---|
| Rossmann | GAT | MAE (↓) | 211 (324) | $252_{\pm21}$ | $304_{\pm18}$ | $\mathbf{230}_{\pm0.55}$ | $289_{\pm5}$ | $564_{\pm59}$ | $963_{\pm40}$ |
| | GATv2 | | 205 (324) | $\mathbf{215}_{\pm9}$ | $330_{\pm11}$ | $239_{\pm2}$ | $251_{\pm8}$ | $531_{\pm91}$ | $1212_{\pm154}$ |
| | GIN | | 178 (324) | $197_{\pm5}$ | $224_{\pm8}$ | $\mathbf{193}_{\pm1}$ | $219_{\pm2}$ | $257_{\pm27}$ | 3648 |
| | G-Conv | | 178 (324) | $206_{\pm9}$ | $229_{\pm1}$ | $\mathbf{192}_{\pm0.74}$ | $216_{\pm4}$ | $253_{\pm28}$ | 3430 |
| | G-SAGE | | 178 (324) | $204_{\pm6}$ | $229_{\pm3}$ | $\mathbf{193}_{\pm0.70}$ | $224_{\pm4}$ | $256_{\pm35}$ | 3428 |
| | RelGNN | | 178 (324) | $210_{\pm11}$ | $223_{\pm12}$ | $\mathbf{192}_{\pm1}$ | $219_{\pm3}$ | $254_{\pm21}$ | 3440 |
| Walmart | GAT | MAE (↓) | 13118 (14.7$k$) | $13626_{\pm147}$ | $13846_{\pm50}$ | $\mathbf{13496}_{\pm67}$ | $13813_{\pm38}$ | $14364_{\pm151}$ | $13795_{\pm76}$ |
| | GATv2 | | 13114 (14.7$k$) | $14081_{\pm291}$ | $13841_{\pm41}$ | $\mathbf{13497}_{\pm35}$ | $13750_{\pm44}$ | $14362_{\pm146}$ | $13760_{\pm163}$ |
| | GIN | | 13056 (14.7$k$) | $15165_{\pm1420}$ | $13841_{\pm19}$ | $\mathbf{13423}_{\pm74}$ | $13755_{\pm71}$ | $14305_{\pm138}$ | $13539_{\pm144}$ |
| | G-Conv | | 13033 (14.7$k$) | $15268_{\pm1540}$ | $13850_{\pm56}$ | $\mathbf{13329}_{\pm23}$ | $13849_{\pm81}$ | $14267_{\pm131}$ | $13581_{\pm103}$ |
| | G-SAGE | | 13052 (14.7$k$) | $15090_{\pm1204}$ | $13838_{\pm43}$ | $\mathbf{13421}_{\pm17}$ | $13738_{\pm43}$ | $14270_{\pm130}$ | $13761_{\pm189}$ |
| | RelGNN | | 13097 (14.7$k$) | $15243_{\pm1682}$ | $13846_{\pm18}$ | $\mathbf{13408}_{\pm29}$ | $13791_{\pm51}$ | $14311_{\pm125}$ | $13722_{\pm161}$ |
| Airbnb | GAT | AUC (↑) | 0.72 (0.5) | $0.64_{\pm0.01}$ | $\mathbf{0.67}_{\pm0.00}$ | $0.64_{\pm0.01}$ | $0.48_{\pm0.01}$ | | $0.46_{\pm0.01}$ |
| | GATv2 | | 0.74 (0.5) | $0.62_{\pm0.01}$ | $\mathbf{0.65}_{\pm0.01}$ | $0.56_{\pm0.07}$ | $0.50_{\pm0.03}$ | | $0.51_{\pm0.00}$ |
| | GIN | | 0.75 (0.5) | $0.62_{\pm0.04}$ | $\mathbf{0.67}_{\pm0.01}$ | $0.66_{\pm0.02}$ | $0.56_{\pm0.01}$ | - | $0.56_{\pm0.01}$ |
| | G-Conv | | 0.74 (0.5) | $0.63_{\pm0.02}$ | $\mathbf{0.69}_{\pm0.02}$ | $0.65_{\pm0.01}$ | $0.54_{\pm0.00}$ | | $0.54_{\pm0.00}$ |
| | G-SAGE | | 0.78 (0.5) | $\underline{0.68}_{\pm0.04}$ | $\mathbf{0.70}_{\pm0.02}$ | $\underline{0.68}_{\pm0.01}$ | $0.63_{\pm0.01}$ | | $0.63_{\pm0.00}$ |
| | RelGNN | | 0.77 (0.5) | $0.66_{\pm0.03}$ | $\mathbf{0.71}_{\pm0.01}$ | $0.67_{\pm0.01}$ | $0.56_{\pm0.03}$ | | $0.60_{\pm0.00}$ |
| Berka | GAT | AUC (↑) | 0.92 (0.5) | $0.56_{\pm0.11}$ | $\underline{0.71}_{\pm0.07}$ | $\mathbf{0.73}_{\pm0.05}$ | | | |
| | GATv2 | | 0.96 (0.5) | $0.44_{\pm0.09}$ | $0.52_{\pm0.08}$ | $\mathbf{0.64}_{\pm0.05}$ | | | |
| | GIN | | 0.96 (0.5) | $\underline{0.61}_{\pm0.05}$ | $0.53_{\pm0.12}$ | $\mathbf{0.68}_{\pm0.13}$ | | - | - |
| | G-Conv | | 0.98 (0.5) | $\mathbf{0.68}_{\pm0.05}$ | $0.35_{\pm0.09}$ | $0.56_{\pm0.15}$ | | | |
| | G-SAGE | | 0.93 (0.5) | $0.56_{\pm0.03}$ | $0.70_{\pm0.12}$ | $\underline{0.61}_{\pm0.14}$ | | | |
| | RelGNN | | 0.95 (0.5) | $\underline{0.55}_{\pm0.17}$ | $0.31_{\pm0.08}$ | $\mathbf{0.65}_{\pm0.15}$ | | | |
| F1 | GAT | AUC (↑) | 0.85 (0.5) | $0.55_{\pm0.05}$ | $\underline{0.66}_{\pm0.04}$ | $\underline{0.66}_{\pm0.04}$ | $\mathbf{0.68}_{\pm0.04}$ | | $0.49_{\pm0.09}$ |
| | GATv2 | | 0.86 (0.5) | $0.73_{\pm0.04}$ | $0.54_{\pm0.10}$ | $\mathbf{0.74}_{\pm0.01}$ | $0.50_{\pm0.15}$ | | $0.56_{\pm0.06}$ |
| | GIN | | 0.87 (0.5) | $\mathbf{0.77}_{\pm0.01}$ | $0.56_{\pm0.05}$ | $\underline{0.76}_{\pm0.02}$ | $0.42_{\pm0.14}$ | | $0.64_{\pm0.12}$ |
| | G-Conv | | 0.87 (0.5) | $\mathbf{0.78}_{\pm0.01}$ | $0.43_{\pm0.11}$ | $0.75_{\pm0.01}$ | $0.46_{\pm0.12}$ | | $0.75_{\pm0.02}$ |
| | G-SAGE | | 0.86 (0.5) | $\mathbf{0.78}_{\pm0.01}$ | $0.44_{\pm0.14}$ | $0.76_{\pm0.02}$ | $0.49_{\pm0.10}$ | | $0.68_{\pm0.05}$ |
| | RelGNN | | 0.84 (0.5) | $\mathbf{0.77}_{\pm0.01}$ | $0.44_{\pm0.08}$ | $0.69_{\pm0.03}$ | $0.46_{\pm0.12}$ | | $0.60_{\pm0.15}$ |

Table 5: **Entity table LightGBM baseline.** We report mean ± SE of ROC-AUC (classification) and MAE (regression). Naive baseline (mean/majority) is in parentheses. "-" indicates invalid time columns. Best is **bold**; within one SE is underlined.

| Dataset | | ORIG. | TARGN | RGCLD | CLAVADDPM | RCTGAN | REALTABF. | SDV |
|---|---|---|---|---|---|---|---|---|
| Rossmann | MAE (↓) | 225 (324) | $227_{\pm0.77}$ | $225_{\pm0.96}$ | $225_{\pm0.06}$ | $\mathbf{218}_{\pm6}$ | $224_{\pm2}$ | $4286_{\pm2}$ |
| Walmart | MAE (↓) | 13664 (14.7$k$) | $13841_{\pm26}$ | $13704_{\pm45}$ | $\mathbf{13666}_{\pm36}$ | $13818_{\pm78}$ | $14433_{\pm129}$ | $13949_{\pm84}$ |
| Airbnb | AUC (↑) | 0.73 (0.5) | $\underline{0.73}_{\pm0.01}$ | $\mathbf{0.74}_{\pm0.01}$ | $0.62_{\pm0.03}$ | $0.71_{\pm0.01}$ | - | $0.51_{\pm0.02}$ |
| Berka | AUC (↑) | 0.71 (0.5) | $0.53_{\pm0.10}$ | $0.62_{\pm0.10}$ | $\mathbf{0.73}_{\pm0.04}$ | - | | - |
| F1 | AUC (↑) | 0.61 (0.5) | $0.53_{\pm0.01}$ | $\mathbf{0.63}_{\pm0.04}$ | $0.57_{\pm0.02}$ | $0.58_{\pm0.04}$ | | $\underline{0.60}_{\pm0.01}$ |

Table 6: **DFS feature engineering LightGBM baseline.** We report mean ± SE of ROC-AUC (classification) and MAE (regression). Naive baseline (mean/majority) is in parentheses. "-" indicates invalid time columns. Best is **bold**; within one SE is underlined.

| Dataset | | ORIG. | TARGN | RGCLD | CLAVADDPM | RCTGAN | REALTABF. | SDV |
|---|---|---|---|---|---|---|---|---|
| Rossmann | MAE (↓) | 100 (324) | $215_{\pm5}$ | $210_{\pm7}$ | $\mathbf{203}_{\pm3}$ | $210_{\pm2}$ | $230_{\pm5}$ | $2290_{\pm67}$ |
| Walmart | MAE (↓) | 13251 (14.7$k$) | $13826_{\pm21}$ | $14560_{\pm830}$ | $\mathbf{13392}_{\pm48}$ | $13741_{\pm32}$ | $14130_{\pm127}$ | $13486_{\pm63}$ |
| Airbnb | AUC (↑) | 0.74 (0.5) | $\underline{0.72}_{\pm0.02}$ | $\mathbf{0.73}_{\pm0.01}$ | $0.60_{\pm0.06}$ | $0.71_{\pm0.02}$ | - | $0.54_{\pm0.01}$ |
| Berka | AUC (↑) | 0.90 (0.5) | $0.64_{\pm0.08}$ | $0.65_{\pm0.05}$ | $\mathbf{0.71}_{\pm0.01}$ | - | | - |
| F1 | AUC (↑) | 0.74 (0.5) | $0.45_{\pm0.01}$ | $\mathbf{0.72}_{\pm0.03}$ | $0.68_{\pm0.01}$ | $0.65_{\pm0.08}$ | | $0.51_{\pm0.02}$ |

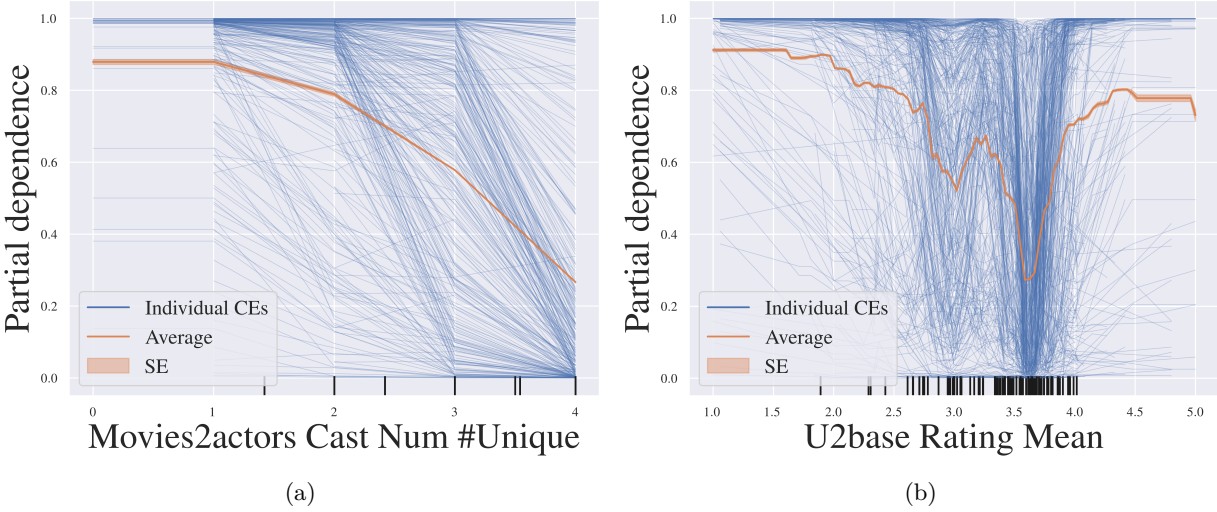

Figure 4: **Partial dependence plots of C2ST**. Results are for the 1st and 4th most important feature from Figure 3. With ideally generated synthetic data, features would not differ between synthetic and original data and every partial dependence plot would be a horizontal line at 50% probability. We can observe that (a) the synthetic data have too many unique actor cast numbers (higher probability of being synthetic when feature value is larger than 4) and (b) the mean movie ratings in the original data vary more than in the synthetic data, where they are more concentrated around 3.5.

However, recent studies have highlighted limitations of distance-based privacy metrics (Yao et al., 2025; Ganev & De Cristofaro, 2025; Ward et al., 2024), showing that such measures can fail to capture information leakage in high-dimensional complex datasets and adversarial scenarios. To provide a more robust and comprehensive evaluation, we therefore complement DCR with attack-based privacy assessment via membership inference attacks (MIA).

In an MIA, an adversary attempts to infer whether a given record from the real data was part of the training set used to generate the synthetic data (El Emam et al., 2022b). We adopt the implementation of Lautrup et al. (2024), which assumes a black-box adversary with knowledge of the population but no insight into the synthesis model. The attacker is modeled as a LightGBM classifier (Ke et al., 2017) trained to distinguish between synthetic samples and real records from an external holdout set. The classifier's discriminative performance serves as an estimate of membership disclosure risk. Analogous to the DCR evaluation, MIA is conducted at the table level, as no multi-table variant currently exists.

We report DCR results in Table 7 and MIA AUC scores in Table 8. Across both tables, all methods—except for ClavaDDPM on the Users table—maintain reasonable DCR values close to the ideal 50% range, suggesting no systematic data copying or excessive proximity between synthetic and real samples. However, the MIA evaluation tells a different story. As shown in Table 8, most generators expose the training data to a non-negligible membership inference risk. With the exception of SDV and ReaLTabFormer, whose MIA AUC scores remain close to random guessing ($\approx 50\%$) but correspond to low data quality (see Tables 2, 3, 4), the adversarial classifier achieves above-random discrimination for nearly all methods. We include detailed results of our privacy experiments in Appendix C.1.

Our privacy evaluation shows that despite not achieving perfect fidelity nor systematically copying the original data, current methods still pose measurable privacy risks to their training data. These findings further confirm recent results highlighting the limitations of distance-based privacy metrics and underscore the importance of attack-based evaluations such as MIA. More broadly, they motivate future research into privacy evaluation protocols for synthetic relational data and the development of privacy-preserving generative models that balance fidelity and privacy protection more effectively (Izzo et al., 2024; Jiang et al., 2025).

Table 7: **DCR Scores** on the Airbnb dataset represent the probability of a random sample being closer to a training sample rather than a randomly drawn sample from a held-out dataset. Values near 50% indicate no systematic data copying, while larger values indicate privacy risks.

| Method | Sessions | Users |
|---|---|---|
| TARGN | 50.40 $_{\pm 0.20}$ | 49.87 $_{\pm 0.50}$ |
| RGCLD | 51.77 $_{\pm 0.21}$ | 50.30 $_{\pm 0.50}$ |
| ClavaDDPM | 51.77 $_{\pm 0.21}$ | 84.83 $_{\pm 0.36}$ |
| RCTGAN | 49.48 $_{\pm 0.22}$ | 48.93 $_{\pm 0.50}$ |
| REALTABF. | 46.65 $_{\pm 1.56}$ | 51.38 $_{\pm 0.50}$ |
| SDV | 46.16 $_{\pm 0.09}$ | 48.01 $_{\pm 0.50}$ |
| SMOTE | 88.86 $_{\pm 0.10}$ | 99.82 $_{\pm 0.04}$ |

Table 8: **MIA AUC Scores** on the Airbnb dataset measure the overall ability of an attacker to distinguish between training records (members) and held-out records (non-members). Values near 50% indicate the attack is equivalent to random guessing, suggesting high privacy protection.

| Method | Sessions | Users |
|---|---|---|
| TARGN | 55.18 $_{\pm 0.10}$ | 59.77 $_{\pm 0.36}$ |
| RGCLD | 50.94 $_{\pm 0.06}$ | 78.52 $_{\pm 0.72}$ |
| CLAVADDPM | 64.43 $_{\pm 0.25}$ | 68.88 $_{\pm 0.32}$ |
| RCTGAN | 62.35 $_{\pm 0.62}$ | 98.76 $_{\pm 0.18}$ |
| REALTABF. | 50.98 $_{\pm 0.52}$ | 50.11 $_{\pm 0.04}$ |
| SDV | 50.00 $_{\pm 0.00}$ | 50.02 $_{\pm 0.01}$ |
| SMOTE | 94.03 $_{\pm 0.11}$ | 92.35 $_{\pm 0.15}$ |

## 5 Conclusion

To address the need for standardized evaluation in the emerging field of synthetic relational database generation, the primary contribution of this work is SYNTHERELA, the first open-source benchmark specifically designed for this task. In developing SYNTHERELA, we critically reviewed existing methods for evaluating the fidelity and utility of synthetic data and introduced two novel contributions: C2ST-Agg, a robust detection-based metric for assessing multi-table fidelity, and RDL-utility, a general framework for evaluating the utility of synthetic relational databases. SYNTHERELA is designed to offer practical insights for synthetic data users and features a public leaderboard to serve as a baseline for future methods. We provide SYNTHERELA as an open-source python package (Appendix B.1).

We also applied the benchmark to current state-of-the-art synthetic data generation methods. The best methods generate marginal distributions well and are on par with single-table methods. Diffusion-based approaches also perform well on single-table fidelity. However, all methods decline in performance when evaluated on multi-table fidelity, highlighting the added complexity of modeling relational databases. Although no method perfectly preserves inter-table relationships, when evaluated on RDL-utility, most outperform naive baselines, and some achieve model performance scores comparable to those trained on the original database. This suggests that despite not achieving perfect fidelity, these synthetic data remain valuable for downstream machine learning tasks.

### 5.1 Limitations

Our benchmark, SYNTHERELA, evaluates neural-based generative methods focused on fidelity and utility on real-world relational databases. This excludes marginal-based methods focused on differential privacy and query error metrics, which are often tailored to specific datasets and queries. The current benchmark datasets lack predefined queries for evaluating these differential privacy-centric approaches. Future work could incorporate datasets with predefined queries and incorporate marginal-based methods to bridge the gap between the two classes of methods.

Several aspects of the evaluation of synthetic data are limited by the difficulty of representative sampling. We circumvent this in RDL-utility by using a temporal-based split; however this limits our current implementation to databases that contain datetime columns. More work needs to be done to understand the limitations and prepare new benchmark datasets or dataset generators.

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

# Appendix

## A   A Survey of Synthetic Relational Database Generation Methods

The **Synthetic Data Vault (SDV)** (Patki et al., 2016) introduced the first learning-based method for generating relational databases. The method is based on the Hierarchical Modeling Algorithm (HMA) synthesizer, which is a multivariate version of the Gaussian Copula method. The method converts all columns to a predefined set of distributions and selects the best-fitting one. To learn dependencies, columns are converted to a standard normal before calculating the covariances. Tables are modeled with a recursive conditional parameter aggregation technique, which incorporates child table covariance and column distribution information into the parent table. The method requires the relational structure or metadata, which has since become a common practice.

The work of Mami et al. (2022) leverages the graph representation of relational database using **Graph Variational Autoencoders**. They focus on the case of one primary table connected by an identifier to an arbitrary number of secondary tables. The approach begins by transforming categorical, datetime, and numeric attributes into a normalised numeric format using an invertible function. Subsequently, all tables' attributes are merged into a single-table, where rows from each table are vertically concatenated. This merged table, along with an adjacency matrix based on foreign key relations, forms a homogeneous graph representation of the dataset. Message passing is then applied to this graph representation using gated recurrent units (GRU). Following the message passing phase, the data is processed through a variational autoencoder, which encodes the joined table and random samples are taken from its latent space. These samples are then decoded back to the data space.

**Composite Generative Models** (Canale et al., 2022) propose a generative framework based on codecs for modeling complex data structures, such as relational databases. They define a codec as a quadruplet: C = (E,D,S,L), consisting of an encoder E producing embeddings and intermediate contexts, a decoder D for distribution representation, a sampler S and loss function L. The authors define the following codecs: Categorical and Numerical Codecs for individual columns, while composite data types are encoded using Struct and List Codecs, allowing for relational database synthesis. They also propose a specific implementation using causal transformers as generative models.

The **Row Conditional-TGAN (RCTGAN)** (Gueye et al., 2023) extends the conditional tabular GAN model (Xu et al., 2019) to relational databases. RCTGAN incorporates data from parent rows into the child table GAN model, allowing it to synthesise data conditionally on the connected parent table rows. The ability for conditional synthesis allows the method to handle various relationship schemas without additional processing. They enhance RCTGAN to capture the influence of grandparent rows on their grandchild rows, preserving this connection even when the relationship information is not transferred by the parent table rows. Database synthesis is based on the row conditional generator of RCTGAN model trained for each table. First, all parent tables are synthesised, followed by sampling the tables for which parents are already sampled. This allows using the synthesised parent rows as features when synthesizing child table rows.

The **Incremental Relational Generator (IRG)** (Li et al., 2024) uses GANs to incrementally fit and sample the relational dataset. They first define a topologically ordered sequence of tables in the dataset. Parent tables are modeled individually, while child tables undergo a three-step generation process. First, a potential context table is constructed by combining data from all related tables through join operations and aggregation. Then, the model predicts the number of child rows to be generated for each parent row, which they call its degree. They then extend the context table with corresponding degrees. Taking this table as context, they use a conditional synthetic tabular data generation model to generate the child table.

The **Realistic Relational and Tabular Transformer (REaLTabFormer)** (Solatorio & Dupriez, 2023) focuses on synthesizing single parent relational data and employs a GPT-2 encoder with a causal language model head to independently model the parent table. The encoder is frozen after training and used to conditionally model the child tables. Each child table requires a new conditional model, implemented as a sequence-to-sequence (Seq2Seq) transformer. The GPT-2 decoder with a causal language model head is trained to synthesise observations from the child table, accommodating arbitrary-length synthetic data

conditioned on an input. While this method supports conditional synthesis of child rows, only one level is supported by this method.

Xu et al. (2023) propose a method for modeling many-to-many (M2M) datasets via random graph generation. They leverage a heterogeneous graph representation of the relational data and propose a factorization for modeling the graph representation incrementally. First, the edges of the graph are generated unconditionally using a random graph model. Second, one of the tables is generated conditionally on the topology of edges. One way to achieve such conditioning is by using a node embedding. Lastly, the remaining tables are generated using the conditional table model, which requires the generation of each node of the table based on the currently generated tables and all connections. They achieve this by using set embeddings to conditionally generate connected tables. The authors propose two variants using different conditional table models **BayesM2M** and **NeuralM2M**.

The **Cluster Latent Variable guided Diffusion Probabilistic Models (ClavaDDPM)** (Pang et al., 2024) utilizes classifier-guided diffusion models, integrating clustering labels as intermediaries between tables connected by foreign-key relations. The authors first propose a model for generating a single parent-child relationship. The connection between the tables is modeled by a latent variable obtained using Gaussian Mixture Model clustering. ClavaDDPM learns a diffusion process on the joint parent and latent variable distribution, followed by training a latent variable classifier on the child table to guide the diffusion model for the child table. Additionally, it includes a model to estimate child group sizes, to preserve relation cardinality. The authors then extend this to more parent-child constraints through bottom-up modeling and address multi-parent scenarios by employing majority voting to mitigate potential clustering inconsistencies.

Hudovernik (2024) adapts the tabular latent diffusion-based model TabSyn (Zhang et al., 2024) for conditional generation of relational databases. The method **Relational Graph-Conditioned Latent Diffusion (RGCLD)** utilizes a heterogeneous graph representation of a relational database. The rows of each table are represented as nodes of a particular type, and the foreign keys between tables are represented by edges connecting the nodes. The method trains a graph neural network for each table to encode the relationships between connected tables. The embeddings of the GNN are used to guide the diffusion process in the latent space. During sampling, the method generates tables sequentially based on a topological order defined by the dataset's schema.

Tiwald et al. (2025) propose **TabularARGN**, an auto-regressive model for generating synthetic data for flat and sequential tables. TabularARGN is a shallow any-order auto-regressive network architecture. The method homogenizes diverse datatypes by discretizing them and is trained by minimizing the categorical cross-entropy loss across the discretized attributes. TabularARGN is designed to generate tabular synthetic data by learning the full set of conditional probabilities across features in a dataset. The sequential table TabularARGN model can handle sequences of arbitrary lengths. TabularARGN's sequential table model can utilize a flat table as context during training and generation, enabling the synthesis of two-table setups, such as a flat table containing time-independent information (e.g., bank customer data) and a sequential table containing time-dependent data (the bank customer transaction histories). The method was open-sourced by the commercial provider Mostly.ai , while the method is specialized for sequential data, they also support generating multi-parent schemas. For these datasets, the method will retain the context for one of the parent tables and retain the referential integrity for the rest[3].

In contrast to neural-based approaches described above, marginal-based approaches focus on preserving low-dimensional measurements of the dataset, such as marginal queries, often under differential privacy (DP), and are typically evaluated by the accuracy of synthesized marginal queries. While marginal-based methods for single-table synthesis are well-established (Zhang et al., 2017; McKenna et al., 2022; Cai et al., 2021), their extension to relational data is a more recent focus. For instance, PrivLava(Cai et al., 2023) achieves DP in relational data through graphical models with latent variables to represent inter-table dependencies. Following this, (Kapenekakis et al., 2024) proposed a method for synthesizing relational data, with a focus on data with sequential aspects like electronic health records. More recently, (Alimohammadi et al., 2025) adapt single-table DP generators to relational data by modeling relationships with a learned bi-adjacency

---

[3] For details on multi-table generation see `https://mostly.ai/docs/generators/configure/set-table-relationships/multi-table`.

matrix, and PrivPetal (Cai et al., 2025) synthesizes DP relational data by modeling a flattened version of the table using permutation relations.

# B  SyntheRela: Synthetic Relational Database Generation Benchmark

We provide our work as a Python package **SYNTHERELA**. The main goal of the package is the evaluation of the quality of synthetic relational databases. We can compare multiple methods across multiple databases with the *Benchmark* class or evaluate a single method on a single database with the *Report* class. All of the results of the benchmark are saved as JSON files and then parsed by our package for results summarization and visualization. The package is open source under the MIT license and can be easily extended with new methods, evaluation metrics, or databases.

## B.1  Using SyntheRela

This section provides a technical guide for users to evaluate synthetic relational data using SYNTHERELA.

**Installation and Setup.** The primary way to interact with the benchmark is through the `syntherela` Python package. It is designed to be highly modular and can be easily installed via the Python Package Index (PyPI): `pip install syntherela`.

**Data Organization and Structure.** Our current implementation assumes that synthetic data is generated in CSV format. It requires a specific directory structure to ensure the `Benchmark` class correctly maps ground-truth relational tables to their synthetic counterparts:

- `data/original/.../`: Contains the ground-truth relational databases. Alongside the csv files the firectory must also contain a `metadata.json` file defining the database schema.

- `data/synthetic/.../method_name/run/sample/`: Contains the synthetic databases for evaluation. We further subdivide by `run` and `sample` subdirectories to account for sample variability.

**The Benchmark Class.** The `Benchmark` class is the central interface of the package. It is compatible with all metrics described in the main paper and provides an extensible framework for users to include their own custom metrics. The class automates the evaluation pipeline, allowing multiple synthesis methods and datasets to be evaluated simultaneously in a single execution. After processing, the benchmark stores all results in a structured JSON format for standardized analysis and leaderboard submission.

**Example.** The following example demonstrates how to programmatically execute the benchmark to evaluate selected methods:

```python
# Import metrics and benchmark class
from syntherela.benchmark import Benchmark
from syntherela.metrics.single_column.statistical import ChiSquareTest
from syntherela.metrics.multi_table.statistical import
    CardinalityShapeSimilarity

# Configuration
benchmark = Benchmark(
    real_data_dir="data/original",
    synthetic_data_dir="data/synthetic",
    results_dir="results",
    single_column_metrics=[ChiSquareTest()],
    multi_table_metrics=[CardinalityShapeSimilarity()],
    datasets=["rossmann_subsampled"],
    methods=["SDV", "RCTGAN"] # Simultaneous evaluation
)

# Run evaluation and generate JSON results
benchmark.run()
```

## B.2 Evaluation Metrics

We list the evaluation metrics for data fidelity and utility currently supported in our benchmark in Table 9, based on the granularity of the data they evaluate. We define three levels of qranularity: *single-column* metrics that evaluate the marginal distributions, *single-table* metrics that evaluate tables, and *multi-table* metrics that evaluate the relational aspects. For single column fidelity, we use the Kolmogorov-Smirnov and $\mathcal{X}^2$ statistical tests, total variation, Hellinger, Jensen-Shannon, and Wasserstein distances, alongside a single column C2ST. We include column pair correlation similarity, maximum mean discrepancy, pairwise correlation difference, and C2ST for single-table fidelity. We evaluate multi-table fidelity with cardinality shape similarity and k-hop correlation similarity alongside C2ST-Agg. We also implement the Parent-Child C2ST (PC-C2ST) for comparison with related work. In our implementation of PC-C2ST, we first split the parent table rows (as in C2ST-Agg) and then apply denormalization. This way we preserve the i.i.d. assumption for the classifier two-sample test, while also including all information from the two tables. Single-column utility is generally covered by fidelity metrics and not evaluated in related work. For single-table utility, we implement tabular machine learning utility metrics, and for multi-table utility, we include RDL-utility.

Table 9: **Evaluation metrics supported in the benchmark.**

|  | **Single Column** | **Single-Table** | **Multi-Table** |
|---|---|---|---|
| **Statistical** | KS Test, $\mathcal{X}^2$ Test | Column pair correlations | cardinality shape similarity |
| **Distance** | Total Variation, Hellinger, Jensen-Shannon, Wasserstein | Maximum Mean Discrepancy, Pairwise Correlation Difference | $k$-hop correlation similarity |
| **Detection** | C2ST | C2ST | C2ST-Agg, PC-C2ST |
| **Utility** | / | Tabular ML-Utility | Relational DL-Utility |

### B.2.1 C2ST with Aggregation

Fidelity methods are concerned with measuring the similarity between two databases with the same schema but different data. Typically, these will be the real database $\mathbb{D}_{\text{REAL}}$ and a synthetic database $\mathbb{D}_{\text{SYN}}$, with the goal of detecting if, to what extent, and where the synthetic data differ from the real data.

Let a relational database $\mathbb{D}$ be a collection of tables $\mathcal{T} = \{T_1, ..., T_n\}$ and a schema $\mathcal{S} = (\mathcal{R}, \mathcal{A})$, where $\mathcal{R} \subseteq \mathcal{T} \times \mathcal{T}$ are the relations between the tables and $A_{T_i} = \{a_1^{T_i}, \ldots, a_l^{T_i}\} \in \mathcal{A}$ define the tables' attributes. Each table is a set $T = \{v_1, ..., v_{n_T}\}$ consisting of elements $v_i$ called rows. Each row $v \in T$ has three components $v = (p_v, \mathcal{K}_v, x_v)$. A **primary key** $p_v$ that uniquely identifies the row $v$; the set of **foreign keys** $\mathcal{K}_v = \{p_{v'} : v' \in T' \text{ and } (T, T') \in \mathcal{R}\}$, where $p_{v'}$ is the primary key of the row $v'$; and the set of **values** $x_v = \{(a, x) : a \in A_T\}$ corresponding to attributes of table $T$.

Algorithm 1 describes how we add aggregations to the target table. For each child table, we add *CountRows*, a count of the number of child rows corresponding to a parent row. For each attribute (i.e. column) in each child table, we compute an aggregation attribute (*mean, count*, etc.). The aggregation attributes are added to the target table. In practice, different aggregation functions may be applied, as long as they maintain the i.i.d. assumption of the data. In our benchmark, we use column means for numerical columns and the number of distinct categories in categorical columns along with the child row count.

Our implementation allows us to directly control how many levels of aggregations we add (similarly to *k*-hop correlation similarity (Pang et al., 2024)). Each level accounts for one application of Algorithm 1. When aggregating for 1 level, only the information directly from the table's children is aggregated; at the second level, the aggregated grandchild columns are also added to the table. In our benchmark, we choose to add only one level of aggregation, as most methods struggle to preserve the relationships at distance $k = 1$.

It has been shown that the accuracy-based approach to two-sample testing is consistent and controls for Type I error and (asymptotically) Type II error (see Kim et al. (2021) for theoretical results and a summary

---

**Algorithm 1 Relational Aggregation.**

---

**Require:** relational database $\mathbb{D}$ with tables $\mathcal{T}$ and relational schema $\mathcal{S} = \{\mathcal{R}, \{A_{T_1} \ldots A_{T_n}\}\}$
**Require:** target table $T$
1: aggregationAttributes ← [ ]
2: **for** each $C_i \in \{C : (C,T) \in \mathcal{R}\}$ **do**
3:      Add $CountRows(C_i, T)$ to aggregationAttributes         ▷ Count number of child rows for each row in $T$.
4:      **for** each $a_j^{C_i} \in A_{C_i}$ **do**         ▷ Iterate through child table columns.
5:          Add $Agg(C_i, a_j^{C_i}, T)$ to aggregationAttributes         ▷ Compute aggregation (e.g., *mean, distinct...*).
6:      **end for**
7: **end for**
8: **for** each $v \in T$ **do**
9:      **for** each **a** ∈ aggregationAttributes **do**
10:          Add (**a**.name, **a**.value) to $v$         ▷ Append computed aggregation attributes.
11:      **end for**
12: **end for**
13: return $T_i$         ▷ Return table with aggregations applied.

---

of empirical results). In practice, we are also interested in finite sample behavior. Experiment-based recommendations show that the approach should have an advantage in power when the data are well-structured or we have a lot of data, or when it is difficult to specify a test statistic, which is very common for high-dimensional data. Therefore, the large, higher-dimensional and structured nature of relational data is a perfect fit for C2ST.

### B.3 Datasets

#### B.3.1 Relational Databases and Sampling Procedures

An important issue with evaluating relational data is that representative sampling is difficult (Buda et al., 2013; Gemulla et al., 2008). If the dataset does not include a time component or if the relationships are non-linear, the sampling becomes non-trivial and directly impacts the performance of the generative method. Even if the data have a strict hierarchy between tables, the rows in a child table are related via their parent, which breaks the assumption of *i.i.d.* sampling. This makes splitting the dataset into a representative train and test set difficult. Consequently, the methods for synthesizing relational databases are typically trained using the entire original dataset.

We organize the datasets used in related work based on the structure of their relational schema. Datasets using only linear relationships (one parent and one child table) include Airbnb (Montoya et al., 2015) and Rossmann Store Sales (FlorianKnauer, 2015). While this structure may be sufficient for some practical applications, Gueye et al. (2023) and Xu et al. (2023) highlight the need for methods supporting more complex, multiple-parent relational structures found in datasets like *MovieLens* (Harper & Konstan, 2015) and *World Development Indicators* (World Bank, 2019). Datasets including multiple child tables include *Telstra Network Disruptions* (Wendy Kan, 2015), *Walmart Recruiting - Store Sales Forecasting* (Walmart, 2014), and *Mutagenesis* (Debnath et al., 1991). Datasets with multiple children and parents include *Coupon Purchase Prediction* (Kato et al., 2015), *World Development Indicators* (World Bank, 2019), *MovieLens* (Harper & Konstan, 2015), *Biodegradability* (Blockeel et al., 1999) and *Berka* Berka et al. (2000).

#### B.3.2 Benchmark Datasets

Table 1 summarizes the relational datasets used in our benchmark. Six datasets are from related work and we add the *Cora* dataset by McCallum et al. (2000), which contains a simple yet challenging relational schema, and **F1** F1 (2021) from the RelBench benchmark. We include 2 datasets per hierarchy type to progressively add complexity in generation. The datasets used in our evaluation are diverse in terms of the number of columns, tables and relationships.

The **Airbnb** (Airbnb, 2015) dataset includes user demographics, web session records, and summary statistics. It provides data about users' interactions with the platform, with the aim of predicting the most likely country of the users' next trip. See Figure 9a for schema.

The **Berka** (also known as the Financial dataset) Berka et al. (2000) dataset contains 606 successful and 76 unsuccessful loans along with their information and transactions. The standard task is to predict the loan outcome for finished loans (A vs B) at the time of the loan start. See Figure 12 for schema.

The **Biodegradability** dataset (Blockeel et al., 1999) comprises a collection of chemical structures, specifically 328 compounds, each labeled with its half-life for aerobic aqueous biodegradation. This dataset is intended for regression analysis, aiming to predict the biodegradation half-live activity based on the chemical features of the compounds. See Figure 13 for schema.

The **Cora** dataset (McCallum et al., 2000) is a widely-used benchmark dataset in the field of graph representation learning. It consists of academic papers from various domains. The dataset consists of 2708 scientific publications classified into one of seven classes and their contents. The citation network consists of 5429 links. See Figure 10b for schema.

The **F1** dataset (F1, 2021) contains Formula 1 racing data and statistics dating back to 1950. It contains information on drivers, constructors, race results, and standings covering every season in F1 history. See Figure 11 for schema.

The **IMDB MovieLens** dataset (Harper & Konstan, 2015) comprises information on movies, actors, directors, and users' film ratings. The dataset consists of seven tables, each containing at least one additional feature besides the primary and foreign keys. See Figure 14 for schema.

The **Rossmann Store Sales** (FlorianKnauer, 2015) features historical sales data for 1115 Rossmann stores. The dataset consists of two tables connected by a single foreign key. This makes it the simplest type of relational dataset. The first table contains general information about the stores and the second contains sales-related data. See Figure 9b for schema.

The **Walmart** dataset (Walmart, 2014) includes historical sales data for 45 Walmart stores across various regions. It includes numerical, date-time and categorical features across three connected tables *store*, *features* and *depts*. The dataset is from a Kaggle competition, with the task of predicting department-wide sales. See Figure 10a for schema.

## B.4 Comparison with Existing Evaluation Tools

Several packages for evaluating synthetic tabular data exist (Nowok et al., 2016; Task et al., 2023; Lautrup et al., 2024; Sidorenko et al., 2025) with the most popular and comprehensive package being Synthcity (Qian et al., 2023a;b). It supports many statistical, privacy and detection-based (with several different models) metrics.

The only package that supports multi-table evaluation is SDMetrics (DataCebo, 2022). It includes multi-table metrics cardinality shape similarity and parent-child detection with logistic detection and support vector classifier. The package is not easy to extend and limits the adaptation of metrics. We re-implement detection metrics (discriminative detection, aggregation detection, and parent-child detection) to be used with an arbitrary classifier supporting the Scikit-learn (Pedregosa et al., 2011; Buitinck et al., 2013) classifier API. In SDMetrics, the results of different metrics are aggregated into a single-value, which limits the comparison of individual metrics between the methods and datasets. We re-implement the distance and statistical metrics so that each statistic, p-value, and confidence interval is easy to access.

Our benchmark package can be easily extended with new methods, metrics, and datasets. The process for adding custom metrics and new datasets is described in `https://github.com/martinjurkovic/syntherela/blob/main/docs/ADDING_A_METRIC.md`.

### B.5 License and Privacy

We obtain the *Airbnb, Biodegradability, Cora, IMDB MovieLens, Rossmann and Walmart* datasets from the public SDV relational demo datasets repository (`https://docs.sdv.dev/sdv/single-table-data/data-preparation/loading-data`, accessed June 6th, 2024.). The SDV project is licensed under the Business Source License 1.1 (`https://github.com/sdv-dev/SDV?tab=License-1-ov-file#readme`), which allows use for research purposes. The Berka dataset was obtained from the CTU Relational Dataset Repository (Motl & Schulte, 2024) an open-access repository of relational databases. The F1 dataset was obtained using the official RelBench implementation Robinson et al. (2024). It is released under the CC-BY-4.0 license (`https://github.com/f1db/f1db?tab=CC-BY-4.0-1-ov-file`). We remove all textual columns from the F1 dataset as relational generative methods are generally unable to generate those. We manually check all of the datasets to ensure they do not include any personally identifiable information. Some of the datasets contain processed columns, including aggregations of numerical values and connected table rows (e.g., nb_rows_in_{related table}). The authors of SDV (Patki et al., 2016) confirmed that these aggregations are not part of the original datasets, so we post-process all of the datasets to include only the columns found in their original form and update the metadata accordingly. We adapt some of the metrics from the SDMetrics (DataCebo, 2022) (MIT License) and Synthcity (Qian et al., 2023a;b) (Apache-2.0 License) synthetic data evaluation tools. We acknowledge Flaticon.com for providing the icons used in Figure 1.

## C  Additional Experiments

### C.1  Privacy Evaluation

In this section, we present additional results from our privacy evaluation experiments, providing a more granular view of the behavior of each generative model. We visualize the full DCR distributions for all evaluated methods, and report the precision and recall metrics of the membership inference attacks.

As shown in Figures 5 and 6, the DCR distributions for most methods exhibit substantial overlap between synthetic–training and synthetic–holdout distances, indicating no systematic data copying. However, ClavaD-DPM on the *users* table shows a noticeable skew toward lower distances between synthetic and training records, consistent with the elevated DCR score reported in Table 7, suggesting potential data copying and privacy issues.

In Tables 11 and 10 we report the *recall* and *precision* of an adversarial classifier, respectively. Recall reflects the probability that rows from the training data are correctly identified as such, and therefore serves as a direct measure of membership disclosure risk. Precision, on the other hand, reflects the probability that a record predicted as a member truly originates from the training set, and serves as a measure of the reliability of the attack's positive predictions (a scores of 50% is analogous to random guessing).

We also examine an additional aspect of privacy: the potential of the C2ST test to detect data copying, similar to the DCR score. To explore this, we conduct a data copying simulation across the benchmark datasets. The experimental setup and a discussion on how common implementations of C2ST metrics may inadvertently mask such privacy risks are provided in Appendix C.4.

### C.2  Single Column Performance

We evaluate the marginal distributions of individual columns by evaluating the column shapes and using the detection metric. Results are summarized in Table 12. TabularARGN performs best on modeling marginal distributions. We hypothesize this is a result of their preprocessing, which removes outliers and discretizes all columns. The method is closely followed by ClavaDDPM, which models individual tables best.

### C.3  Shortcomings of Logistic Detection

As explained in Section 2.2 a significant limitation of LD is its inability to capture interactions between columns. It can thus assign a perfect fidelity score to a dataset that is completely corrupted. In this section, we empirically show this shortcoming. We conduct the experiment by selecting a table from each dataset

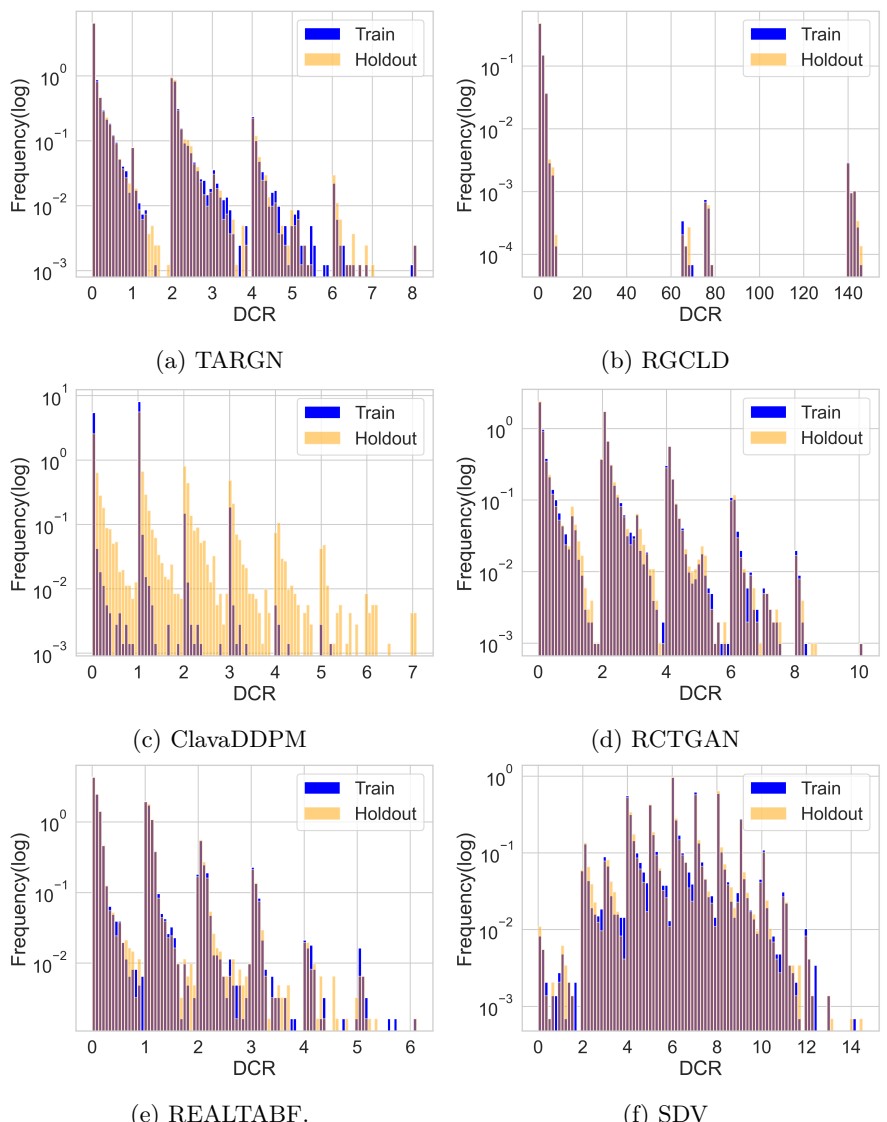

Figure 5: **DCR Distributions** for the *users* table in the anonymized *Airbnb* dataset. The y-axis is log-scaled for a better comparison.

(with the exception of CORA in which no table has two columns, which are not primary or foreign keys). We first select the table and split it in half to simulate the original table and a perfectly generated (by the underlying DGP) synthetic table. We then copy the "generated" table and randomly shuffle values in each column, completely ruining the fidelity of the dataset while keeping the marginal distributions intact. We then evaluate the perfectly generated and shuffled datasets using LD and C2ST using XGBoost. The results are visualized in Figure 7.

Notably LD assigns both versions of the dataset the same score, labeling them indistinctive from the original data. If the fidelity aspect of interest would be solely the marginal distributions, the LD results would be more appropriate than those of C2ST using XGBoost (as marginals are identical in both datasets). However, given that we are interested in single-table fidelity, our experiment showcases a fundamental shortcoming of LD as a measure of single-table fidelity.

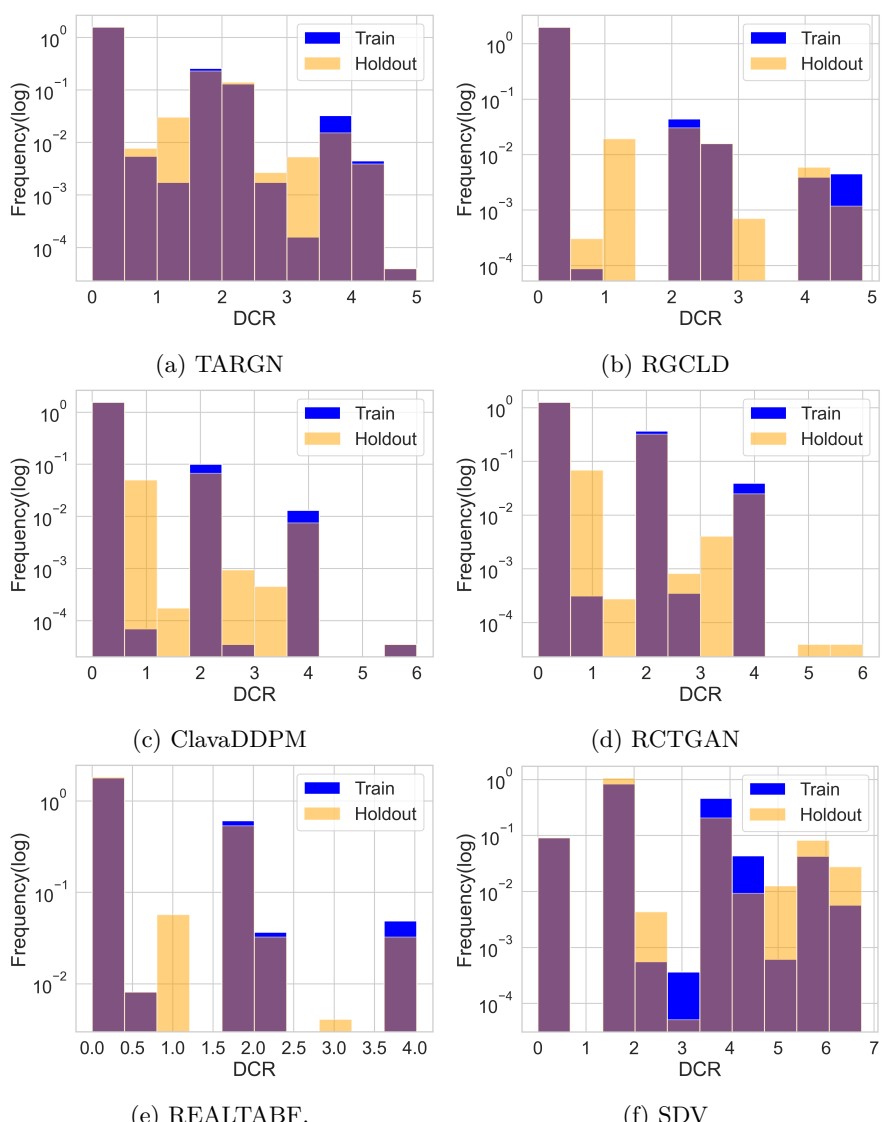

Figure 6: **DCR Distributions** for the *sessions* table in the anonymized *Airbnb* dataset. The *sessions* table consists primarily of categorical columns, resulting in highly discretized DCR distributions. Despite this, most methods maintain a reasonable level of privacy. The y-axis is log-scaled for a better comparison.

## C.4  Classifier Accuracy as a Data Copying Diagnostic

We investigate how the accuracy of the discriminative model in classifier two sample testing can be used to diagnose data copying in Figure 8. We also demonstrate how the classifier performance commonly reported in LD $(2 \cdot \max(\text{AUC}, \frac{1}{2}) - 1)$ masks this issue.

As in the previous experiment, we simulate a perfect synthetic generating a dataset by splitting the original table in half. However, instead of introducing corruption into the second half, we create an exact copy of the original data (i.e., the first half). The commonly used LD implementation fails to detect data copying and assigns the copied data a perfect score. In contrast, C2ST (with XGBoost) accuracy successfully detects data copying as accuracy drops significantly below 50%. We then examine the behaviour of C2ST when only a portion of the data is copied. We keep a portion of the dataset as an identical copy and sample the rest of the values from the "perfectly generated" half. For most of the datasets, even when a relatively low percentage of the data is copied, the model detects the duplication.

Table 10: **MIA Recall** denotes the probability that the attack predicts a record as a member given that it is truly a member of the training set. Values significantly above 50% indicate privacy concerns.

| Method | Sessions | Users |
|--------|----------|-------|
| TARGN | $11.27_{\pm 0.15}$ | $17.58_{\pm 2.06}$ |
| RGCLD | $2.14_{\pm 0.17}$ | $61.31_{\pm 2.21}$ |
| CLAVA | $30.03_{\pm 2.74}$ | $37.49_{\pm 0.71}$ |
| RCTGAN | $28.16_{\pm 0.85}$ | $98.55_{\pm 0.35}$ |
| REALTABF. | $3.14_{\pm 0.38}$ | $0.19_{\pm 0.07}$ |
| SDV | $0.03_{\pm 0.00}$ | $0.03_{\pm 0.02}$ |
| SMOTE | $91.02_{\pm 0.17}$ | $85.24_{\pm 0.34}$ |

Table 11: **MIA Precision** denotes the probability that a record is truly a member of the training set given that the attack predicted it as a member. We estimate uncertainty with standard errors.

| Method | Sessions | Users |
|--------|----------|-------|
| TARGN | $92.14_{\pm 0.48}$ | $99.49_{\pm 0.12}$ |
| RGCLD | $86.04_{\pm 0.58}$ | $99.78_{\pm 0.05}$ |
| CLAVADDPM | $92.81_{\pm 0.71}$ | $99.94_{\pm 0.03}$ |
| RCTGAN | $93.11_{\pm 0.21}$ | $99.72_{\pm 0.09}$ |
| REALTABF. | $70.76_{\pm 6.15}$ | $62.67_{\pm 17.62}$ |
| SDV | $85.00_{\pm 10.00}$ | $35.00_{\pm 21.79}$ |
| SMOTE | $97.14_{\pm 0.18}$ | $99.86_{\pm 0.03}$ |

Table 12: **Single Column Results.** We report the detection accuracy for C2ST (lower is better), and the KS/TV complement for column shapes (higher is better). For each dataset and metric, we report the average across all tables for three independent samples. SDV exceeds the sampling time limit on IMDB (TLE) and "-" denotes a method is unable to generate the dataset. The best result is **in bold** and results within one standard deviation underlined.

| Dataset | Metric | TARGN | RGCLD | ClavaDDPM | RCTGAN | REALTABF. | SDV | MARE |
|---------|--------|-------|-------|-----------|--------|-----------|-----|------|
| Airbnb | C2ST ($\downarrow$) | $\mathbf{51.88}_{\pm 0.09}$ | $52.40_{\pm 0.36}$ | $55.34_{\pm 0.02}$ | $56.37_{\pm 0.02}$ | $61.68_{\pm 0.61}$ | $70.36_{\pm 0.03}$ | $58.57_{\pm 5e\text{-}3}$ |
| | Shapes ($\uparrow$) | $\mathbf{95.70}_{\pm 0.05}$ | $95.57_{\pm 0.63}$ | $94.42_{\pm 0.01}$ | $89.18_{\pm 0.17}$ | $71.66_{\pm 0.92}$ | $59.37_{\pm 0.04}$ | $83.82_{\pm 3e\text{-}3}$ |
| Rossmann | C2ST ($\downarrow$) | $\mathbf{51.07}_{\pm 0.19}$ | $52.76_{\pm 0.33}$ | $54.79_{\pm 0.08}$ | $54.98_{\pm 0.23}$ | $53.62_{\pm 0.17}$ | $59.99_{\pm 0.10}$ | - |
| | Shapes ($\uparrow$) | $\mathbf{96.96}_{\pm 0.19}$ | $95.83_{\pm 0.55}$ | $94.05_{\pm 0.07}$ | $91.31_{\pm 0.04}$ | $90.65_{\pm 0.38}$ | $81.05_{\pm 0.19}$ | - |
| Walmart | C2ST ($\downarrow$) | $59.25_{\pm 0.29}$ | $59.34_{\pm 0.80}$ | $\mathbf{52.60}_{\pm 0.55}$ | $64.12_{\pm 0.40}$ | $59.08_{\pm 0.31}$ | $66.71_{\pm 0.58}$ | - |
| | Shapes ($\uparrow$) | $89.09_{\pm 0.33}$ | $88.43_{\pm 1.00}$ | $\mathbf{92.21}_{\pm 0.52}$ | $82.31_{\pm 0.51}$ | $81.71_{\pm 0.40}$ | $81.80_{\pm 0.10}$ | - |
| Berka | C2ST ($\downarrow$) | $58.64_{\pm 0.26}$ | $60.00_{\pm 1.30}$ | $\mathbf{47.25}_{\pm 0.12}$ | $57.35_{\pm 0.20}$ | - | $70.77_{\pm 0.17}$ | - |
| | Shapes ($\uparrow$) | $82.20_{\pm 0.26}$ | $78.91_{\pm 2.42}$ | $\mathbf{91.62}_{\pm 0.10}$ | $81.90_{\pm 0.38}$ | - | $56.27_{\pm 0.29}$ | - |
| F1 | C2ST ($\downarrow$) | $61.61_{\pm 0.04}$ | $\mathbf{57.69}_{\pm 0.67}$ | $60.76_{\pm 0.20}$ | $63.71_{\pm 0.22}$ | - | $79.85_{\pm 0.22}$ | - |
| | Shapes ($\uparrow$) | $84.71_{\pm 1.15}$ | $\mathbf{91.04}_{\pm 1.51}$ | $84.63_{\pm 0.28}$ | $\underline{89.68}_{\pm 0.40}$ | - | $52.62_{\pm 0.57}$ | - |
| IMDB | C2ST ($\downarrow$) | $50.36_{\pm 0.12}$ | $53.02_{\pm 1.50}$ | $\mathbf{49.86}_{\pm 0.06}$ | $53.74_{\pm 0.08}$ | - | - | - |
| | Shapes ($\uparrow$) | $98.40_{\pm 0.14}$ | $93.61_{\pm 2.75}$ | $\mathbf{99.01}_{\pm 0.05}$ | $92.70_{\pm 0.09}$ | - | - | - |
| Biodegradability | C2ST ($\downarrow$) | $\mathbf{55.42}_{\pm 0.48}$ | $61.85_{\pm 1.57}$ | - | $57.84_{\pm 0.40}$ | - | $64.24_{\pm 0.37}$ | - |
| | Shapes ($\uparrow$) | $\underline{90.85}_{\pm 0.14}$ | $84.53_{\pm 3.42}$ | - | $\mathbf{90.91}_{\pm 0.40}$ | - | $79.46_{\pm 0.47}$ | - |
| Cora | C2ST ($\downarrow$) | $50.94_{\pm 0.37}$ | $52.98_{\pm 1.03}$ | - | $\mathbf{48.97}_{\pm 0.14}$ | - | $75.45_{\pm 0.16}$ | - |
| | Shapes ($\uparrow$) | $92.65_{\pm 0.12}$ | $88.27_{\pm 1.94}$ | - | $\mathbf{96.38}_{\pm 0.15}$ | - | $50.24_{\pm 0.17}$ | - |

# D Relational Deep Learning Utility

We adopt the RelBench (Robinson et al., 2024) relational deep learning benchmark to support synthetic data into the relational deep learning utility framework. It is designed to support training GNN models both on real-world and synthetic relational datasets, and evaluating on a real held-out test set, in a *train-on-synthetic-evaluate-on-real* paradigm. RDL-utility allows integration of new databases and models.

The framework creates two separate dataset-task pairs. One pair includes the synthetic dataset with training and validation splits used for model training. The other pair contains the real dataset with an additional test split used for final evaluation. Both pairs follow the same schema but are kept separate to avoid data leakage. Graph construction converts relational tables into heterogeneous graphs for both synthetic and real datasets. The process maps tables to node types and foreign key relations to edge types. Temporal splits are applied consistently to ensure only past data is available during training and testing. GNN models are trained with temporal awareness, restricting access to data strictly before the test time.

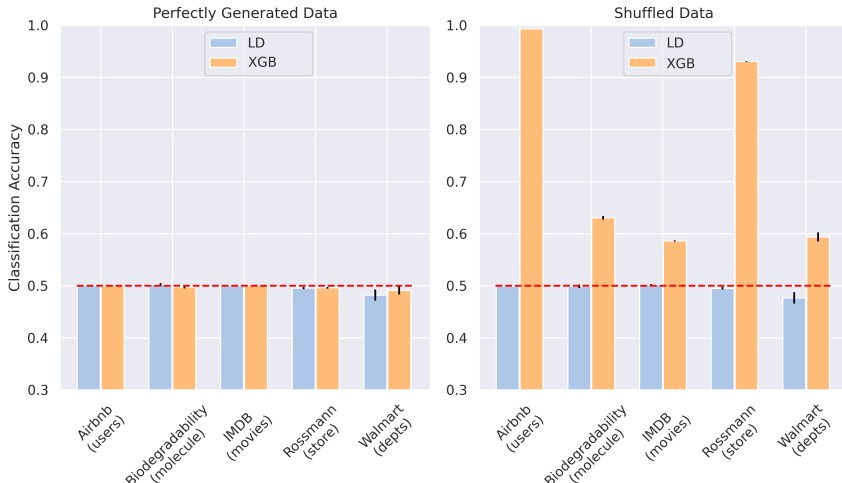

Figure 7: **Issues with logistic detection.** For each dataset, we simulate a perfectly generated table by splitting the original table in half. We copy one part of the table and shuffle the values in each column and thus completely ruin the fidelity of the table. While the XGBoost classifier can almost perfectly segment the corrupted rows, logistic regression assigns both of the datasets the same score.

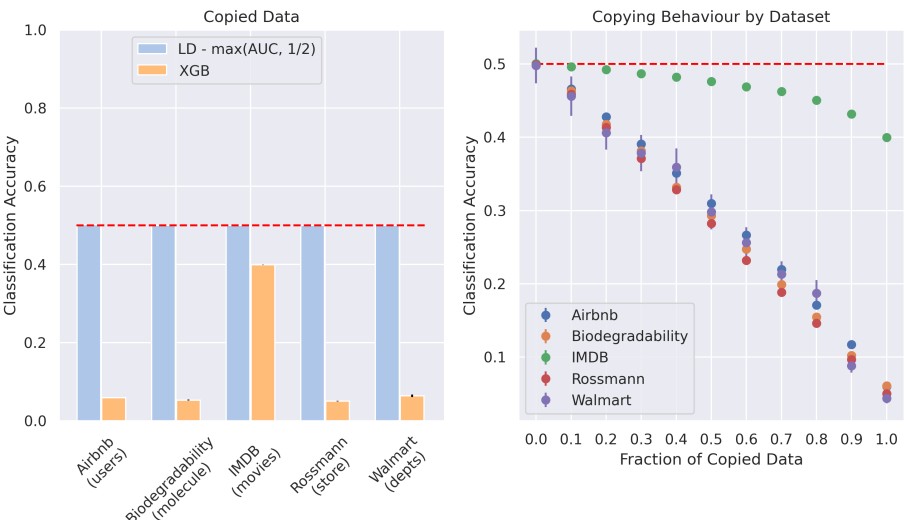

Figure 8: **Detecting data copying with C2ST.** The left plot demonstrates how the error estimation of LD $(2 \cdot \max(\text{AUC}, \frac{1}{2}) - 1)$ masks data copying, while C2ST with XGBoost detects it across all datasets. In the right plot, we observe how copying only a fraction of the original data affects discriminative model accuracy, with accuracy consistently decreasing as more data is duplicated.

## D.1 Utility Tasks

The summary of the RDL-utility tasks is displayed in Table 13. We define the AutoComplete task for the Rossmann, Airbnb, Walmart and Berka datasets based on the predefined tasks by the dataset authors from Kaggle competitions or other dataset information. The task objectives per datasets are as defined as follows:

- **Rossmann** [Regression]: We predict the daily number of customers for each store.

Table 13: **Task statistics for relational datasets used in evaluation.**

| Dataset | Task name | Task type | #Rows of Task Table | | | #Unique Entities | %Train-Val/Test Entity Overlap |
|---|---|---|---|---|---|---|---|
| | | | Train | Validation | Test | | |
| rossmann | Autocomplete | Regression | 46,750 | 9,350 | 28,985 | 46,750 | 0.0 |
| Walmart | Autocomplete | Regression | 5,925 | 2,952 | 11,946 | 5,925 | 0.0 |
| Airbnb | Autocomplete | Classification | 1,661 | 217 | 999 | 1,661 | 0.0 |
| berka | Autocomplete | Classification | 328 | 196 | 158 | 328 | 0.0 |
| f1 | Driver-top3 | Classification | 1,353 | 588 | 726 | 92 | 50.0 |

- **Walmart** [Regression]: We predict the weekly sales for each department of each store.

- **Airbnb** [Classification]: We predict whether a user has already made a booking or not.

- **Berka** [Classification]: We predict the binary status of the loan (successful or unsuccessful).

- **F1** [Classification]: We predict whether a driver will qualify in the top-3 for a race in the next month, which is one of the original RelBench predictive tasks for this dataset.

## D.2 Hyperparameter Tuning

As GNN models are sensitive to hyperparameters Tönshoff et al. (2024), we perform an exhaustive hyperparameter search across all datasets and GNN variants. We use the `Optuna` library Akiba et al. (2019) to create a hyperparameter search for each dataset and GNN variant pair on the original data. We run 100 experiments for each dataset-GNN pair and optimize the parameters specified in Table 14.

Table 14: **Hyperparameter search specification for GNN training.** We run 100 experiments for each dataset-GNN variant pair.

| Hyperparameter | Type | Search Space / Values |
|---|---|---|
| Learning rate | Categorical | {0.0001, 0.001, 0.01, 0.1, 0.5, 1.0} |
| Number of layers | Categorical | {1, 2, 3} |
| Number of neighbors | Categorical | {-1, 128} |
| Weight decay | Continuous (log scale) | [1e-9, 0.01] |
| Number of MLP layers | Categorical | {1, 2, 3} |
| Aggregation method | Categorical | {sum, mean, max, min} |

The best hyperparameters used in RDL-utility are specified in the Reproducibility section of the Appendix in Table 15.

## D.3 Utility Baselines

To assess the effectiveness of RDL-utility in quantifying the utility performance of synthetic relational databases, we implement two baseline approaches that measure utility on single tables.

The single-table baseline evaluates utility using only the entity table. As the predictive model, we use LightGBM Ke et al. (2017), a gradient boosting decision tree implementation. The model is trained with ten iterations of automated hyperparameter tuning on the training and validation splits, and the best configuration is evaluated on the test set.

The second baseline implements the automated feature generation deep feature synthesis (Kanter & Veeramachaneni, 2015). Here, relational structure is incorporated through automated feature generation: for categorical columns, we apply `count` aggregation; for numerical columns, we use `mean` and `sum`; and for datetime columns, we extract the `month` and `year`. Aggregations are applied recursively throughout the database to produce a flattened feature table. This table is then used to train LightGBM with the same

Table 15: **RDL-utility hyperparameter specification.** We list the best hyperparameters from the hyperparameter search of 100 runs on original data.

| GNN | Dataset | LR | Layers | Neighbors | Weight Decay | MLP Layers | Aggr |
|---|---|---|---|---|---|---|---|
| Hetero-GAT | Rossmann | 0.001 | 2 | 128 | 5.4e-05 | 1 | sum |
| | Walmart | 1.0 | 2 | 128 | 2.6e-06 | 1 | sum |
| | Airbnb | 0.0001 | 1 | 128 | 2.7e-07 | 1 | min |
| | Berka | 0.01 | 2 | 128 | 1.3e-07 | 2 | mean |
| | F1 | 0.5 | 1 | 128 | 6.4e-05 | 2 | mean |
| Hetero-GATv2 | Rossmann | 0.001 | 2 | 128 | 2.1e-08 | 3 | sum |
| | Walmart | 0.5 | 2 | 128 | 1.6e-07 | 2 | max |
| | Airbnb | 0.0001 | 1 | -1 | 1.8e-09 | 2 | max |
| | Berka | 0.1 | 2 | 128 | 8.7e-06 | 2 | min |
| | F1 | 0.1 | 1 | -1 | 3.5e-07 | 3 | sum |
| Hetero-GIN | Rossmann | 1.0 | 2 | -1 | 1.3e-05 | 1 | sum |
| | Walmart | 0.1 | 2 | 128 | 1.3e-08 | 2 | mean |
| | Airbnb | 0.001 | 1 | -1 | 3.4e-04 | 2 | max |
| | Berka | 0.1 | 2 | 128 | 1.6e-05 | 2 | max |
| | F1 | 0.5 | 3 | -1 | 1.5e-06 | 1 | mean |
| Hetero-GraphConv | Rossmann | 0.5 | 1 | 128 | 1.0e-04 | 3 | min |
| | Walmart | 1.0 | 2 | 128 | 2.5e-08 | 2 | sum |
| | Airbnb | 0.001 | 1 | -1 | 3.0e-04 | 1 | sum |
| | Berka | 0.01 | 2 | 128 | 2.3e-09 | 3 | max |
| | F1 | 0.1 | 2 | 128 | 9.9e-07 | 3 | sum |
| Hetero-GraphSAGE | Rossmann | 0.1 | 1 | -1 | 0.00152 | 3 | sum |
| | Walmart | 0.5 | 3 | 128 | 2.1e-09 | 2 | mean |
| | Airbnb | 0.001 | 2 | -1 | 8.5e-07 | 1 | max |
| | Berka | 0.001 | 3 | -1 | 8.5e-07 | 3 | max |
| | F1 | 0.1 | 1 | 128 | 4.0e-07 | 3 | sum |
| RelGNN | Rossmann | 1.0 | 1 | 128 | 2.3e-05 | 3 | sum |
| | Walmart | 0.1 | 2 | 128 | 5.2e-05 | 2 | sum |
| | Airbnb | 0.001 | 2 | -1 | 4.4e-06 | 1 | sum |
| | Berka | 0.01 | 2 | 128 | 5.6e-06 | 2 | min |
| | F1 | 0.01 | 2 | -1 | 6.5e-05 | 1 | mean |

procedure as in the single-table baseline, allowing a direct comparison between learning from the entity table alone and learning with additional relational features.

# E    Experimental Setup

## E.1    Computational Resources

The generative methods were trained on NVIDIA 32GB V100S GPUs and H100 80GB GPUs. The total number of GPU hours spent across all experiments is approximately 500. Results which do not require a GPU were run on machines running AMD EPYC 7702P 64-Core Processor with 256GB of RAM. All experiments were performed on an internal HPC cluster.

## E.2    Reproducibility

### E.2.1    Datasets and Data Splitting

Scripts for downloading the datasets and their metadata in the SDV format (Patki et al., 2016) are available in the project repository, as well as the corresponding synthetic data samples for all methods to enable the reproduction of the benchmark results.

We opt not to split the datasets into train, test, and validation sets for generative model training. When no temporal information is included and the structure is non-linear the representative sampling in relational datasets is non-trivial. We delegate this to future work.

Due to computational limits (also reported by Solatorio & Dupriez (2023)), we subsample the Rossmann Store Sales, Airbnb, and Walmart datasets. Additionally, we subsample the Berka and F1 datasets for the purposes of obtaining a held-out test set for our utility experiments.

Our choice to subsample these datasets may limit the generality of our findings to longer time horizons. As the evaluated methods are primarily deep-learning methods, they should generally achieve better performance given more data. However, we apply a fixed subsampling procedure and the same data is used for all methods. Meaning that our results remain comparable within our benchmark and should be interpreted as method comparisons under a consistent computational budget. We also provide full versions of these databases, fully compatible with our benchmark.

- **Rossmann Store Sales**: Subsampled on table *historical*, column *Date* by taking the rows of a two month period from *2014-07-31* to *2014-09-30*, similarly to Solatorio & Dupriez (2023).

- **Airbnb**: Subsampled the dataset by only including the users who have less than 50 sessions and then sampled 10k users, as done by Solatorio & Dupriez (2023).

- **Walmart**: Subsampled on tables *departments* and *features* on the column *Date* by taking the rows from January 2012.

- **Berka**: We use the data prior to *1998-01-01* and use the *1998* data as the test set.

- **F1**: We use the data prior to *2010-01-01* for training. This corresponds to the test set timestamp in the official RelBench implementation [4].

### E.2.2 Experimental Details and Hyperparameters

To provide some quantification of the variability from the non-deterministic nature of the methods, we generated synthetic data for each of the methods for each of the datasets 3 times with different fixed random seeds. We ran the benchmark for each replication.

Scripts for reproducing the generative model training and instructions for training commercial methods are included in the project repository.

It is possible that better performance could be achieved by investing more effort into parameter tuning. However, due to our choice to not split the data, it was not clear how to optimize hyperparameters; therefore, we selected default hyperparameters for all methods (see Tables 16 and 17).

## F   Broader impacts

The research introduces a benchmark for evaluating the quality of synthetic relational databases. This could improve the development of synthetic data, which may be beneficial in fields with privacy restrictions, for example, healthcare (Appenzeller et al., 2022), finance (Assefa et al., 2020), education (Bonnéry et al., 2019), and in cases of limited or biased data (Ntoutsi et al., 2020; Rajpurkar et al., 2022). However, there are possible negative aspects. Synthesis processes might have weaknesses that affect privacy and amplify biases already present in the original data. Also, synthetic data that resembles real data could be misused. Therefore, further work on privacy protection, bias reduction, and detection of synthetic data is needed for its proper use.

---

[4]https://relbench.stanford.edu/datasets/rel-f1/

Table 16: **Hyperparameter specification** for RCTGAN (Gueye et al., 2023), SDV (Patki et al., 2016) and RealTabFormer Solatorio & Dupriez (2023).

| model | hyperparameter | value |
|---|---|---|
| RCTGAN | embedding__dim | 128 |
| | generator__dim | (256, 256) |
| | discriminator__dim | (256, 256) |
| | generator__lr | 0.0002 |
| | generator__decay | 1e-06 |
| | discriminator__lr | 0.0002 |
| | discriminator__decay | 1e-06 |
| | batch_size | 500 |
| | discriminator__steps | 1 |
| | epochs | 1000 |
| | pac | 10 |
| | grand__parent | True |
| | field_transformers | None |
| | constraints | None |
| | rounding | "auto" |
| | min__value | "auto" |
| | max__value | "auto" |
| SDV | locales | None |
| | verbose | True |
| | table_synthesizer | "GaussianCopulaSynthesizer" |
| | enforce__min__max__values | True |
| | enforce_rounding | True |
| | numerical_distributions | {} |
| | default_distribution | "beta" |
| REALTABFORMER | epochs | 100 |
| | batch__size | 8 |
| | train_size | 0.95 |
| | output__max__length | 512 |
| | early__stopping__patience | 5 |
| | early__stopping__threshold | 0 |
| | mask__rate | 0 |
| | numeric__nparts | 1 |
| | numeric__precision | 4 |
| | numeric__max__len | 10 |
| | evaluation__strategy | "steps" |
| | metric__for__best__model | "loss" |
| | gradient__accumulation__steps | 4 |
| | remove__unused__columns | True |
| | logging__steps | 100 |
| | save__steps | 100 |
| | eval__steps | 100 |
| | load__best__model__at__end | True |
| | save__total__limit | 6 |
| | optim | "adamw__torch" |

Table 17: **Hyperparameter specification** for TabularARGN (Tiwald et al., 2025), ClavaDDPM (Pang et al., 2024) and RGCLD Hudovernik (2024).

| model | hyperparameter | value |
|---|---|---|
| TARGN (API) | Configuration presets | Accuracy |
| | Max sample size | 100% |
| | Model size | Large |
| | Batch size | Auto |
| | Flexible generation | Off |
| | Value protection | Off |
| CLAVADDPM | num_clusters | 50 |
| | parent_scale | 1.0 |
| | classifier_scale | 1.0 |
| | num_timesteps | 2000 |
| | batch_size | 4096 |
| | layers_diffusion | [512, 1024, 1024, 1024, 1024, 512] |
| | iterations_diffusion | 200000 |
| | lr_diffusion | 0.0006 |
| | weight_decay_diffusion | 1e-05 |
| | scheduler_diffusion | "cosine" |
| | layers_classifier | [128, 256, 512, 1024, 512, 256, 128] |
| | iterations_classifier | 20000 |
| | lr_classifier | 0.0001 |
| | dim_t | 128 |
| RGCLD | GNN hidden dim | 128 |
| | GNN aggregation | sum |
| | GNN layers | # Tables |
| | GNN lr | 0.008 |
| | GNN weight decay | $1e-5$ |
| | GNN epochs | 250 |
| | VAE layers | 2 |
| | VAE token dim | 4 |
| | VAE hidden dim | 128 |
| | VAE $\delta$ | 0.7 |
| | VAE $(\beta_{max}, \beta_{min})$ | $(0.01, 1e-5)$ |
| | VAE lr | $1e-3$ |
| | VAE epochs | 4000 |
| | Diff model dim | 1024 |
| | Diff lr | 0.001 |
| | Diff weight decay | $1e-6$ |
| | Diff epochs | 4000 |

## G  Dataset Schema

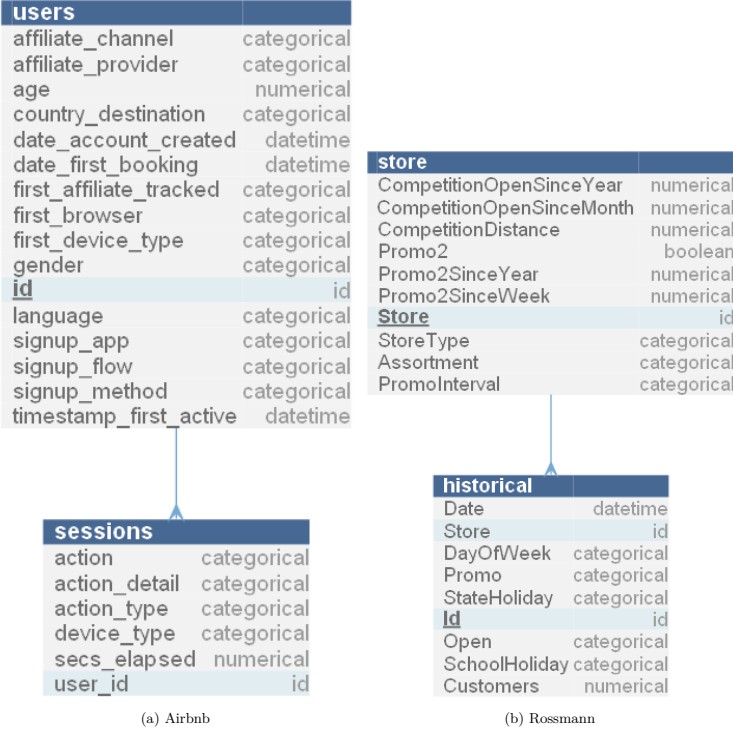

(a) Airbnb                      (b) Rossmann

Figure 9: Two table database diagrams.

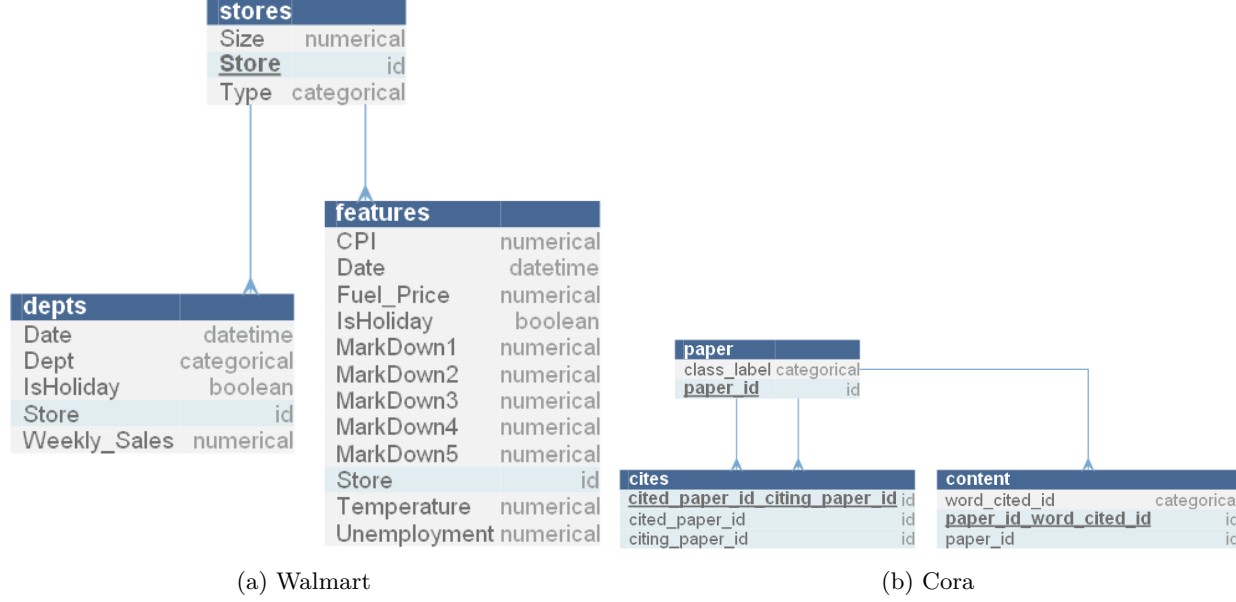

(a) Walmart                              (b) Cora

Figure 10: Walmart and CORA database diagrams.

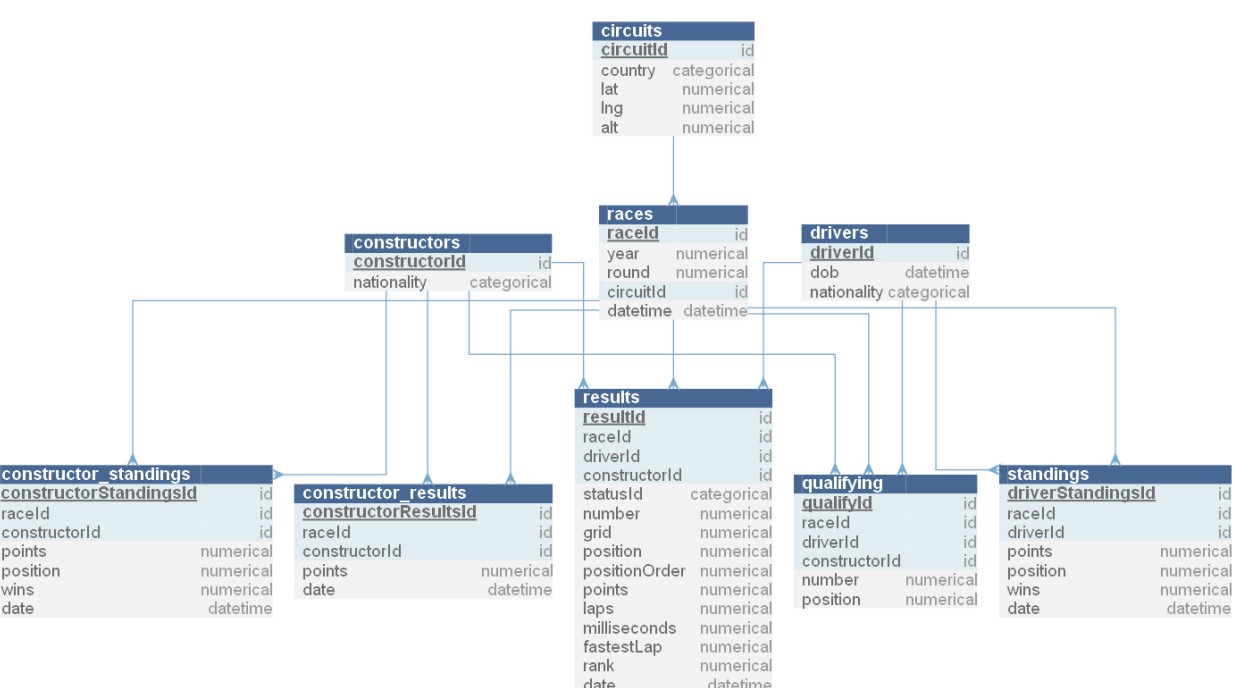

Figure 11: F1 database diagram.

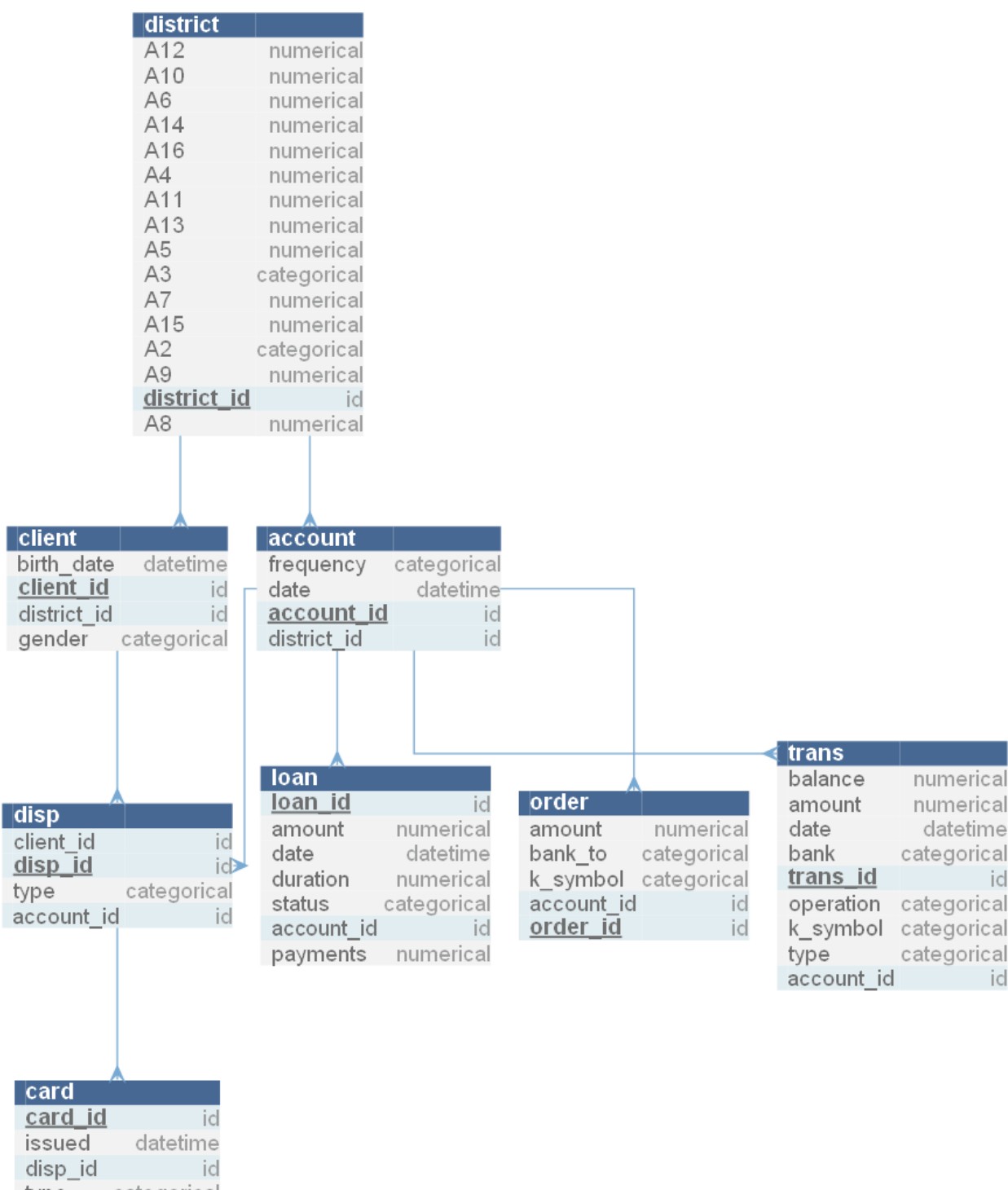

Figure 12: Berka database diagram.

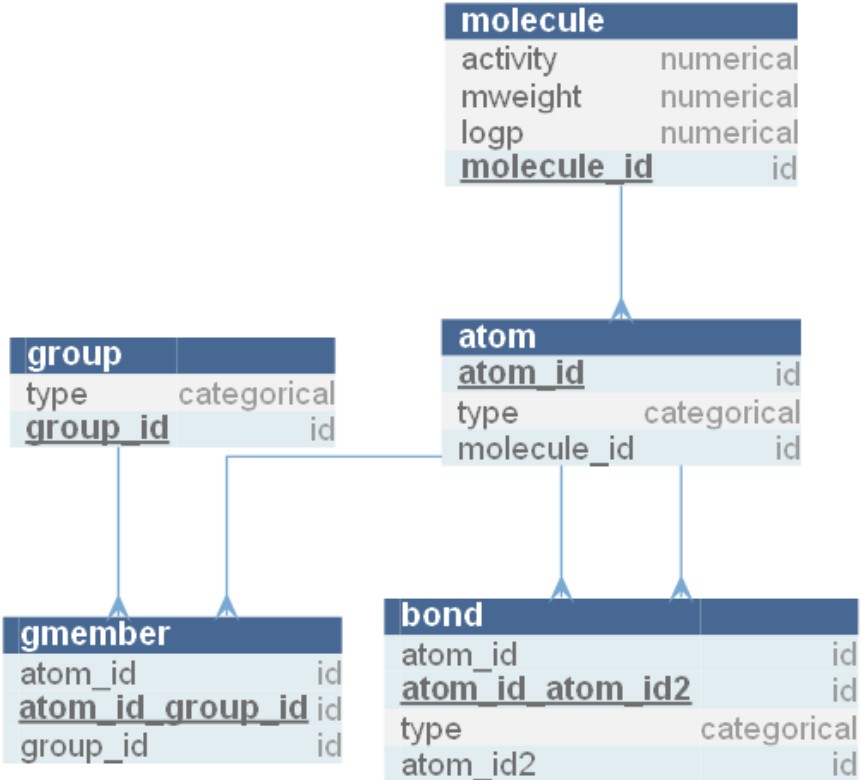

Figure 13: Biodegradability database diagram.

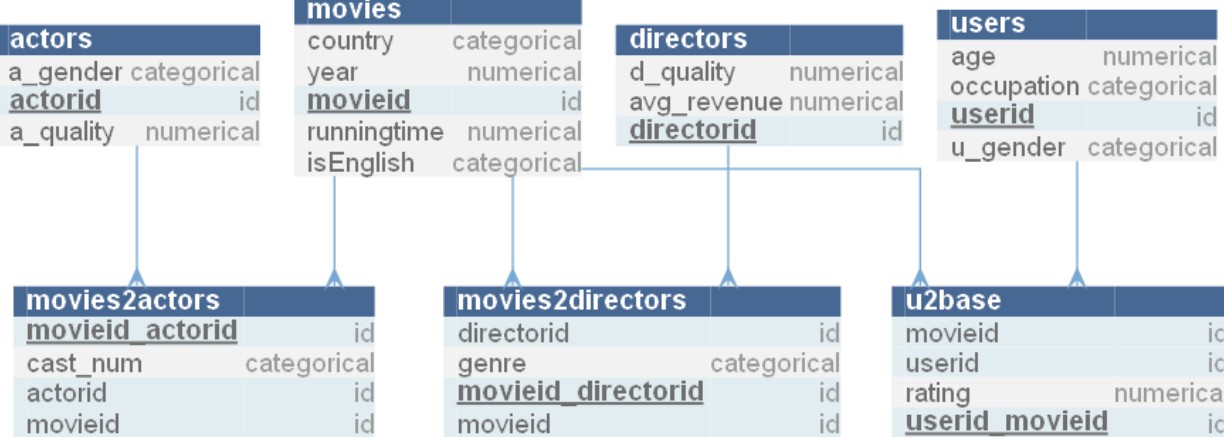

Figure 14: IMDB MovieLens database diagram.

