# OpenReview forum: "SyntheRela: A Benchmark For Synthetic Relational Database Generation"
_TMLR — Accepted by TMLR_

### Review · Reviewer_98gm · 2025-11-17

**Summary Of Contributions:**

The authors propose a new benchmark suite for evaluating generative models for relational data. The authors propose two new metrics: classifier two-sample test with aggregation (C2ST-Agg) and relational deep learning utility (RDL utility). Along with these new metrics, the authors evaluate 6 methods on 8 datasets including various characteristics, including single and multi table generation. Thus, the authors benchmark the applicable models in a broad setting and various evaluation methods.

While the authors suggest that their benchmark suite is well suited for other methods, it remains a bit vague why certain choices have been made (e.g., why the 8 chosen datasets).

**Additional Comments:**

The statement is confusing:
> We include 6 datasets that feature in related work (Airbnb, Rossmann, Walmart, Biodegradability, MovieLens)

Sometimes it is unclear why abbreviations are introduced. E.g.,  "machine learning efficacy (ML-E)"

**Audience:**

Yes

**Audience Explanation:**

The field of generating relational tables seems to be emerging and surely such works benefit from systematic evaluation procedures in literature.

**Broader Impact Concerns:**

No concerns

**Claims And Evidence:**

Yes

**Claims Explanation:**

While I am not an expert on generative models of relational data, the chosen dimensions that are being evaluated appear reasonable to me and align well with how I intuitively would structure such a benchmark.

**Requested Changes:**

The impact of such benchmarks is largely driven by the usability of the code. However, the readme is rather scarce on how to add a new model. The authors should spend more time on providing simple exaplanations etc. on what needs to be done.

---

> ### Author Response · Authors · 2025-12-31
>
> We would like to thank the reviewer for their time and feedback. We are pleased that the reviewer found our contribution worthwhile and the evaluation dimensions to be reasonable and well-aligned with a principled approach to benchmarking synthetic data generation.
>
> ---
>
> #### **[W1] Dataset Selection and Rationale**
> > *While the authors suggest that their benchmark suite is well suited for other methods, it remains a bit vague why certain choices have been made (e.g., why the 8 chosen datasets). …Confusing statement: We include 6 datasets that feature in related work (Airbnb, Rossmann, Walmart, Biodegradability, MovieLens)*
>
> **Response:**
> We have clarified the rationale behind our dataset selection and updated the confusing statement in the manuscript.
>
> **Action Taken:** We revised **Section 4** to explicitly state that we chose six datasets (Airbnb, Rossmann, Walmart, Biodegradability, MovieLens, and IMDB) because they are used in existing literature. We added the **Cora** dataset due to its status as a benchmark in the graph research community and introduced the **F1** dataset, borrowed from **RelBench**, to expand our selection of predictive tasks for RDL-Utility.
>
> #### **[W2] Use of Abbreviations**
> > *Sometimes it is unclear why abbreviations are introduced. E.g., "machine learning efficacy (ML-E)"*
>
> **Response:**
> We agree that unused abbreviations can hinder the readability of the manuscript. We removed all unused abbreviations, including "ML-E."
>
> #### **[W3] Code Usability and Documentation**
> > *The impact of such benchmarks is largely driven by the usability of the code. However, the readme is rather scarce on how to add a new model. The authors should spend more time on providing simple explanations etc. on what needs to be done.*
>
> **Response:**
> We thank the reviewer for reviewing our codebase. Improving the usability and documentation of the benchmark is a priority for us to ensure wider adoption.
>
> **Action Taken:**
> 1. We added a **Jupyter notebook example** providing a step-by-step guide on how to use the benchmark and referenced it in the README.
> 2. We added a dedicated section on using the benchmark to both the **README** and **Appendix B**.
> 3. Based on responses from **Reviewer cyMs** we also include a guide for submitting to the leaderboard.
> Alongside these changes, the README also includes references to documentation on **how to add a new metric** and **how to replicate the paper's results**, including links to download all datasets and the corresponding results for all generative methods evaluated in the study.
>
> ---
> We trust that the updated documentation and the clarified dataset rationale improve the usability and transparency of the benchmark. We are happy to engage in further dialogue if any points remain unclear.

---

### Review · Reviewer_9amB · 2025-11-23

**Summary Of Contributions:**

The paper introduces SyntheRela, a benchmark for evaluating synthetic relational database generators. It contributes: 1. C2ST-Agg, a new detection-based metric for multi-table fidelity. 2. RDL-Utility, a relational deep learning utility evaluation using GNNs. 3. A comprehensive empirical comparison across 6 models and 8 real-world relational datasets. Code and leaderboard are provided. The benchmark addresses a gap in evaluating full relational synthetic data, going beyond single-table methods.

**Audience:**

Yes

**Audience Explanation:**

Relational synthetic data is an active research area and the community lacks established evaluation tools. A benchmark that evaluates multi-table fidelity and relational utility would interest researchers working on synthetic data, relational learning and data-centric AI. The topic is relevant to a meaningful part of the TMLR audience.

**Claims And Evidence:**

Yes

**Claims Explanation:**

The paper’s main claims are supported by clear experiments using six generative models and eight relational datasets. The proposed C2ST-Agg metric and RDL-Utility are validated through consistent empirical results, diagnostics and comparisons to existing metrics. The benchmark is well documented and limitations are acknowledged. The evidence is sufficient to support the contributions.

**Requested Changes:**

These are not essential for acceptance but would strengthen the paper:
1. Clarify how dataset subsampling may affect the generality of the results.
2. Add a short note explaining when RDL-Utility is applicable since it requires temporal structure.
3. Provide a short discussion on how differential privacy oriented methods could be incorporated in future versions of the benchmark.
4. Give one simple example of relational errors that C2ST-Agg may not detect to help readers understand its scope.

---

> ### Author Response · Authors · 2025-12-31
>
> We thank the reviewer for their time and constructive feedback. We appreciate that the reviewer finds the topic of our paper relevant to the TMLR audience and considers our benchmark and empirical comparison to be comprehensive.
>
> ---
>
> #### **[W1] Impact of Dataset Subsampling**
> > *Clarify how dataset subsampling may affect the generality of the results.*
>
> **Response:**
> Our decision to use subsampled versions of certain datasets was primarily to account for the computational constraints of established methods at the time (such as RealTabFormer). While subsampled datasets might not fully represent the original data distribution (e.g., limiting the temporal range), we use a fixed subsampling procedure and the same data across all generative methods. This ensures the results remain comparable within our benchmark and represent a method comparison under a consistent computational budget.
>
> **Action Taken:**
> 1. We added a paragraph commenting on the effect of subsampling on the results.
> 2. We clarified that as newer methods offer more scalable generation, we also include the full versions of all datasets, which are fully compatible with our benchmark.
>
> #### **[W2] Applicability of RDL-Utility**
> > *Add a short note explaining when RDL-Utility is applicable since it requires temporal structure.*
>
> **Response:**
> We agree that the dependency on temporal structure is a key constraint that should be made explicit to the reader.
>
> **Action Taken:**
> 1. We updated our note on RDL-utility in the **Limitations** section to make it clear that it is only applicable to databases that contain datetime columns.
> 2. Additionally, we clarified this requirement in **Section 3.2**.
>
> #### **[W3] Incorporating differential privacy oriented methods**
> > *Provide a short discussion on how differential privacy oriented methods could be incorporated in future versions of the benchmark.*
>
> **Response:**
> In principle, our benchmark already supports including DP-oriented or marginal-based methods, as it can be applied to any method that produces a synthetic version of a database based on the provided schema. The current reason for excluding these methods was due to missing implementations [1] or implementations tailored to certain datasets, that would require significant manual modification to generalize.  We see unifying marginal-based and neural network-based methods as a promising future research direction, as was done for tabular generators [2]. However, at present, existing marginal-based implementations do not allow for synthesizing databases of arbitrary schemas without significant per-database modifications.
>
> **Action Taken:** We clarified the benchmark's compatibility with newer methods in the **“Benchmark Usage”** section added to **Appendix B**.
>
>
> #### **[W4] Scope and Limitations of C2ST-Agg**
> > *Give one simple example of relational errors that C2ST-Agg may not detect to help readers understand its scope.*
>
> **Response:**
> A current limitation of C2ST-Agg is tied to the choice of aggregation functions used during evaluation. We provide a diagnostic example below.
>
> Consider a scenario where a parent table $T1$ is linked 1:N to a child table $T2$ containing a numerical column $C1$. If a generator perfectly reproduces the features of $T1$, the conditional mean of $C1$, and the counts of connected rows, but fails to capture the marginal distribution of $C1$, C2ST-Agg (with current mean/count aggregations) will assign the generative method a perfect score.
>
> Adding further aggregations or utilizing single column C2ST in addition to C2ST-Agg would be necessary to reveal the poorly generated data. However, including additional simple, automated relational features [3] as used in our baselines for utility evaluation (Table 6) can circumvent this limitation and make C2ST-Agg an even stricter metric of multi-table fidelity.
>
> ---
>
> We thank the reviewer again for these constructive suggestions, which we believe have strengthened the paper. We hope our responses clarify the scope of the benchmark and look forward to any additional feedback.
>
> ---
>
> [1] Xu, Kai, et al. "Synthetic data generation of many-to-many datasets via random graph generation." *The Eleventh International Conference on Learning Representations*, 2022.
> [2] Ganev, Georgi, Kai Xu, and Emiliano De Cristofaro. "Graphical vs. deep generative models: Measuring the impact of differentially private mechanisms and budgets on utility." *Proceedings of the 2024 ACM SIGSAC Conference on Computer and Communications Security*, 2024.
> [3] Kanter, James Max, and Kalyan Veeramachaneni. "Deep feature synthesis: Towards automating data science endeavors." *2015 IEEE international conference on data science and advanced analytics (DSAA)*. IEEE, 2015.

---

### Review · Reviewer_cyMs · 2025-12-16

**Summary Of Contributions:**

### Summary
The authors present a benchmarking framework for evaluating methods that generate synthetic relational databases. The evaluation spans multiple scenarios, including single-table and multi-table fidelity analysis, utility evaluation, and privacy risk assessment. The authors compare several generative approaches and report error bars for all results. In addition, a qualitative analysis using selected examples (Figures 3 and 4) illustrates how the benchmark can help reveal characteristic failure modes of existing methods.

### Strengths:
* (S1 – relevance): Relational data synthesis is practically important, making the problem both relevant and timely for the community.
* (S2 – clarity of positioning): The paper clearly motivates why relational data synthesis is challenging and why a dedicated benchmark is necessary.
* (S3 – experiments, results, and metrics): The authors conduct a comprehensive experimental evaluation covering data fidelity, downstream utility, and privacy risk. Reporting error bars strengthens the significance of the results, and the chosen metrics appear reasonable and well motivated.

### Weaknesses:
* (W1 – narrative flow, clarity, and presentation):
While the overall motivation becomes clear after reading the full manuscript, the narrative flow feels fragmented at several points. Restructuring and language polishing would improve readability. For example:
    - Before reviewing generative methods, it would be helpful to first introduce the core evaluation criteria for synthetic data quality (e.g., fidelity, utility, and privacy). In particular, differential privacy (DP) is used later without being clearly introduced.
     - In Section 2.2, the discussion of statistical, distance-based, and detection-based fidelity metrics is well structured, but the preceding paragraph on data granularity appears misplaced and interrupts the flow.
    - Although the paper proposes a benchmark, it remains unclear which components are part of the final benchmarked framework and how they are intended to be used in practice (see W4).

* (W2 – metric sensitivity / minor):
The benchmark relies in part on classifier-based fidelity metrics. While effective, such metrics can be sensitive to the choice of discriminator and may be susceptible to models exploiting weaknesses of the scorer rather than genuinely improving data quality. This is a minor concern and does not affect the overall contribution, but a brief discussion of this limitation would strengthen the paper.

* (W3 – limited citation support for fidelity claims):
In Section 3.1, the authors argue that existing state-of-the-art fidelity metrics fail to adequately capture inter-table relationships. While this claim would benefit from stronger support through explicit references or a more detailed discussion of prior limitations.

* (W4 – guidance for practitioners):
Although the comparative evaluation is thorough, the manuscript provides limited guidance on how practitioners are expected to use the benchmark. For instance, it is unclear whether users interact with SyntheRela via a Python package, how generative models or results are submitted to the leaderboard, and what level of automation is provided. Clarifying these aspects would improve usability and adoption.

**Audience:**

Yes

**Audience Explanation:**

Data scarcity is a relevant problem; therefore, generative approaches for synthetic data and corresponding benchmarks are important. This also applies to relational data.

**Broader Impact Concerns:**

No concerns.

**Claims And Evidence:**

No

**Claims Explanation:**

Generally yes, only (W3) should be fixed.

**Requested Changes:**

### Critical
* (W3): The claim that existing fidelity metrics fail to adequately capture inter-table relationships should be supported with additional literature references and a more detailed discussion, as this claim directly motivates the introduction and use of the proposed C2ST metric.

### Not critical but recommended
* (W1): The manuscript would benefit from further polishing to improve narrative flow and readability, as some sections currently feel fragmented.
* (W2): A short discussion of the potential risk that generative models may exploit weaknesses of classifier-based fidelity metrics (rather than improving true data quality) would strengthen the evaluation.
* (W4): Providing clearer guidance on how the framework is intended to be used in practice (e.g., workflow, tooling, or interaction with the leaderboard) would improve usability for practitioners.

---

> ### Author Response · Authors · 2025-12-31
>
> We thank the reviewer for their thoughtful and constructive feedback. We are encouraged that the reviewer found our contribution worthwhile, the benchmark well structured, and the evaluation metrics reasonable and well motivated.
>
> Below, we address the specific concerns raised, particularly the requested critical changes regarding citation support of existing multi-table metrics.
>
> ---
>
> ### **General Revisions**
> In response to the feedback from all reviewers, we made several updates to our manuscript. The key changes include:
> * **Restructuring Section 2** to provide a clearer introduction to core evaluation criteria (fidelity, utility, privacy) and improve narrative flow.
> * **Extending Appendix B** to provide a "Practitioner’s Guide" for the `syntherela` Python package.
> * **Strengthening Section 3.1** with references to existing literature regarding multi-table metrics.
> ---
>
> #### **[W1] Narrative flow, clarity, and presentation**
> > *While the overall motivation becomes clear after reading the full manuscript, the narrative flow feels fragmented at several points. Restructuring and language polishing would improve readability.*
>
> **Response:**
> We agree that a high-level overview of evaluation criteria helps ground the subsequent discussion of methods.
>
> > *Before reviewing generative methods, it would be helpful to first introduce the core evaluation criteria for synthetic data quality (e.g., fidelity, utility, and privacy). In particular, differential privacy (DP) is used later without being clearly introduced.*
>
> **Action Taken:**
> 1. We added an introductory paragraph to **Section 2** outlining the three aspects of synthetic data evaluation (fidelity, utility, and privacy). Within this section, we now formally introduce differential privacy and include additional references to related work.
> > *In Section 2.2, the discussion of statistical, distance-based, and detection-based fidelity metrics is well structured, but the preceding paragraph on data granularity appears misplaced and interrupts the flow.*
> 2. To resolve the fragmentation in **Section 2.2**, we moved the technical paragraph on data granularity to **Appendix B.2**, where we provide a comprehensive breakdown of metrics by both type and granularity.
>
> #### **[W2] Metric sensitivity of classifier-based metrics**
> > *The benchmark relies in part on classifier-based fidelity metrics. While effective, such metrics can be sensitive to the choice of discriminator and may be susceptible to models exploiting weaknesses of the scorer rather than genuinely improving data quality.*
>
> **Response:**
> The reviewer raises an important point regarding the use of weak discriminators for detection-based fidelity metrics. We believe that a key issue of previous work was a lenient fidelity evaluation due to utilizing a logistic regression discriminator. For this reason we advocate for (Section 3.1) and use (Section 4) strong, tree-based boosting methods which have been shown to be strong discriminators by previous work on tabular data synthesis [1].
>
> #### **[W3] Limited citation support for fidelity claims (Critical)**
> Answered in follow-up comment.
>
> #### **[W4] Guidance for practitioners**
> > *The manuscript provides limited guidance on how practitioners are expected to use the benchmark. For instance, it is unclear whether users interact with SyntheRela via a Python package, how generative models or results are submitted to the leaderboard, and what level of automation is provided.*
>
> **Response:**
> Improving the usability of the benchmark is a priority for us.
>
> **Action Taken:**
> 1. We now explicitly state in **Section 5** that SyntheRela is available as a Python package.
> 2. We significantly expanded **Appendix B**, adding a section that details the workflow of the `syntherela` package, including data preparation, metric selection, and benchmark usage.
> 3. **Leaderboard Guidelines:** Following the reviewer's suggestion to look at established benchmarks [2], we have added comprehensive [submission guidelines](https://docs.google.com/document/d/1ae16L_vvT5PFt2OeN7FJauA_ayd_A6xCkhVJFoYcx04/edit?tab=t.0) to our project repository's README to clarify the process for future leaderboard participants.
>
> ---
> [1] Zein, EL Hacen, and Tanguy Urvoy. "Tabular Data Generation: Can We Fool XGBoost?." *NeurIPS 2022 First Table Representation Workshop*, 2022.
> [2] Li, Jinyang, et al. "Can LLM already serve as a database interface? A big bench for large-scale database grounded text-to-SQLs." *Advances in Neural Information Processing Systems 36*, 2023.

---

> ### Author Response · Authors · 2025-12-31
> **On W3 (Additional support for multi-table fidelity claims)**
>
> > *The claim that existing state-of-the-art fidelity metrics fail to adequately capture inter-table relationships should be supported with additional literature references and a more detailed discussion.*
>
> **Response:**
> We have adjusted our claim by specifying that the aspect these metrics fail to evaluate is the interactions between tables induced by the foreign key relationships. Cardinality similarity, k-hop similarity, and classifier-based fidelity using denormalization are the only three metrics used in related work for multi-table fidelity.
>
> **Action Taken:**
> 1. We updated **Section 3.1** with references to the metrics used in prior work.
> 2. We expanded our discussion on the shortcomings of these metrics
> The stated limitations of cardinality (that it only accounts for relationship cardinality) and k-hop similarity (only accounting for interactions between pairs of columns in related tables) are evident from how the methods are defined.
> 3. We also expanded upon the issues of parent-child logistic detection. The first obvious issue is the use of logistic regression as the discriminator, which can be easily addressed. However, the second, more nuanced issue is the fact that denormalization breaks the i.i.d. assumption for C2ST.  First, we note that the commonly used implementation of denormalization is flawed: denormalizing before randomly splitting into train and test allows rows sharing the same parent to appear in both sets, which can artificially inflate the discriminative model’s capacity and unfairly penalize generative methods. Our aggregation-based method avoids this issue by splitting at the parent level. To address this issue, we include a corrected version of the parent-child detection metric in our evaluation package. We first split the parent table rows (as in C2ST-Agg) and then apply the denormalization, preserving the i.i.d nature of the data while also accounting for cross-table interactions. To account for this, we updated our description in Appendix B.2.
>
> ---
>
> We hope our answer addresses the reviewers' concerns and remain open to further discussion to ensure the clarity of our claims meets TMLR standards.

---

> > ### Comment · Reviewer_cyMs · 2026-01-03
> > **Answer to the authors**
> >
> > Dear authors,
> >
> > Thank you for the clear description of the manuscript updates (responses, action items, and highlighted changes). With these updates, the issues I previously raised have been addressed, and I will vote for acceptance in the recommendation section.

---

### Decision · Action_Editor_n3G7 · 2026-01-30

**Recommendation:** Accept with minor revision

**Additional Comments:**

All new discussions in the rebuttal are expected to be incorporated into the camera-ready version.

**Audience:**

Yes

**Audience Explanation:**

This work provides a good benchmark for evaluating synthetic relational data, which is practically important.

**Claims And Evidence:**

Yes

**Claims Explanation:**

The submission has explained why relational data synthesis is challenging and why a dedicated benchmark is necessary.